# 🦙 Swift: On-the-Fly Self-Speculative Decoding for LLM Inference Acceleration

**Heming Xia**[1], **Yongqi Li**[1],[*] **Jun Zhang**[2], **Cunxiao Du**[3], **Wenjie Li**[1]

[1]Department of Computing, The Hong Kong Polytechnic University
[2]College of Computer Science and Technology, Zhejiang University    [3]Sea AI Lab
{he-ming.xia}@connect.polyu.hk; {zj.cs}@zju.edu.cn

## Abstract

Speculative decoding (SD) has emerged as a widely used paradigm to accelerate LLM inference without compromising quality. It works by first employing a compact model to draft multiple tokens efficiently and then using the target LLM to verify them in parallel. While this technique has achieved notable speedups, most existing approaches necessitate either additional parameters or extensive training to construct effective draft models, thereby restricting their applicability across different LLMs and tasks. To address this limitation, we explore a novel *plug-and-play* SD solution with layer-skipping, which skips intermediate layers of the target LLM as the compact draft model. Our analysis reveals that LLMs exhibit great potential for self-acceleration through layer sparsity and the task-specific nature of this sparsity. Building on these insights, we introduce Swift, an on-the-fly self-speculative decoding algorithm that adaptively selects intermediate layers of LLMs to skip during inference. Swift does not require auxiliary models or additional training, making it a *plug-and-play* solution for accelerating LLM inference across diverse input data streams. Our extensive experiments across a wide range of models and downstream tasks demonstrate that Swift can achieve over a $1.3\times\sim1.6\times$ speedup while preserving the original distribution of the generated text. We release our code in https://github.com/hemingkx/SWIFT.

## 1 Introduction

Large Language Models (LLMs) have exhibited outstanding capabilities in handling various downstream tasks (OpenAI, 2023; Touvron et al., 2023a;b; Dubey et al., 2024). However, their token-by-token generation necessitated by autoregressive decoding poses efficiency challenges, particularly as model sizes increase. To address this, *speculative decoding* (SD) has been proposed as a promising solution for lossless LLM inference acceleration (Xia et al., 2023; Leviathan et al., 2023; Chen et al., 2023). At each decoding step, SD first employs a compact draft model to efficiently predict multiple tokens as speculations for future decoding steps of the target LLM. These tokens are then validated by the target LLM in parallel, ensuring that the original output distribution remains unchanged.

Recent advancements in SD have pushed the boundaries of the *latency-accuracy* trade-off by exploring various strategies (Xia et al., 2024), including incorporating lightweight draft modules into LLMs (Cai et al., 2024; Ankner et al., 2024; Li et al., 2024a;b), employing fine-tuning strategies to facilitate efficient LLM drafting (Kou et al., 2024; Yi et al., 2024; Elhoushi et al., 2024), and aligning draft models with the target LLM (Liu et al., 2023a; Zhou et al., 2024; Miao et al., 2024). Despite their promising efficacy, these approaches require additional modules or extensive training, which limits their broad applicability across different model types and causes significant inconvenience in practice. To tackle this issue, another line of research has proposed the *Jacobi-based drafting* (Santilli et al., 2023; Fu et al., 2024) to facilitate *plug-and-play* SD. As illustrated in Figure 1(a), these methods append pseudo tokens to the input prompt, enabling the target LLM to generate multiple tokens as drafts in a single decoding step. However, the Jacobi-decoding paradigm misaligns with the autoregressive pretraining objective of LLMs, resulting in suboptimal acceleration effects.

---

[*]Corresponding Author

In this work, we introduce a novel research direction for plug-and-play SD: *sparsity-based drafting*, which leverages the inherent sparsity in LLMs to enable efficient drafting (see Figure 1(b)). Specifically, we exploit a straightforward yet practical form of LLM sparsity – *layer sparsity* – to accelerate inference. Our approach is based on two key observations: **1) LLMs possess great potential for self-acceleration through layer sparsity.** Contrary to the conventional belief that layer selection must be carefully optimized (Zhang et al., 2024), we surprisingly found that uniformly skipping layers to draft can still achieve a notable $1.2\times$ speedup, providing a strong foundation for plug-and-play SD. **2) Layer sparsity is task-specific.** We observed that each task requires its own optimal set of skipped layers, and applying the same layer configuration across different tasks would cause substantial performance degradation. For example, the speedup drops from $1.47\times$ to $1.01\times$ when transferring the configuration optimized for a storytelling task to a reasoning task.

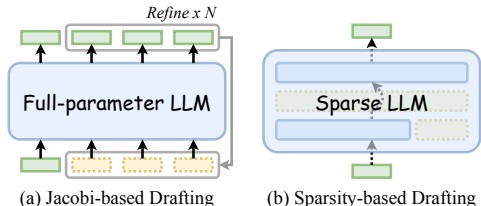

(a) Jacobi-based Drafting     (b) Sparsity-based Drafting

Figure 1: Illustration of prior solution and ours for *plug-and-play* SD. (a) Jacobi-based drafting appends multiple pseudo tokens to the input prompt, enabling the target LLM to generate multiple tokens as drafts in a single step. (b) SWIFT adopts sparsity-based drafting, which exploits the inherent sparsity in LLMs to facilitate efficient drafting. This work is the first exploration of plug-and-play SD using sparsity-based drafting.

Building on these observations, we introduce SWIFT, the first *on-the-fly* self-speculative decoding algorithm that adaptively optimizes the set of skipped layers in the target LLM during inference, facilitating the lossless acceleration of LLMs across diverse input data streams. SWIFT integrates two key innovations: (1) a *context-based* layer set optimization mechanism that leverages LLM-generated context to efficiently identify the optimal set of skipped layers corresponding to the current input stream, and (2) a *confidence-aware* inference acceleration strategy that maximizes the use of draft tokens, improving both speculation accuracy and verification efficiency. These innovations allow SWIFT to strike an expected balance between the *latency-accuracy* trade-off in SD, providing a new *plug-and-play* solution for lossless LLM inference acceleration *without the need for auxiliary models or additional training*, as demonstrated in Table 1.

We conduct experiments using LLaMA-2 and CodeLLaMA models across multiple tasks, including summarization, code generation, mathematical reasoning, etc. SWIFT achieves a $1.3\times\sim1.6\times$ wall-clock time speedup compared to conventional autoregressive decoding. Notably, in the greedy setting, SWIFT consistently maintains a $98\%\sim100\%$ token acceptance rate across the LLaMA2 series, indicating the high alignment potential of this paradigm. Further analysis validated the effectiveness of SWIFT across diverse data streams and its compatibility with various LLM backbones.

Our key contributions are:

(i) We performed an empirical analysis of LLM acceleration on layer sparsity, revealing both the potential for LLM self-acceleration via layer sparsity and its task-specific nature, underscoring the necessity for adaptive self-speculative decoding during inference.

(ii) Building on these insights, we introduce SWIFT, the first *plug-and-play* self-speculative decoding algorithm that optimizes the set of skipped layers in the target LLM on the fly, enabling lossless acceleration of LLM inference across diverse input data streams.

(iii) We conducted extensive experiments across various models and tasks, demonstrating that SWIFT consistently achieves a $1.3\times\sim1.6\times$ speedup without any auxiliary model or training, while theoretically guaranteeing the preservation of the generated text's distribution.

## 2 RELATED WORK

**Speculative Decoding (SD)** Due to the sequential nature of autoregressive decoding, LLM inference is constrained by memory-bound computations (Patterson, 2004; Shazeer, 2019), with the primary latency bottleneck arising not from arithmetic computations but from memory reads/writes of LLM parameters (Pope et al., 2023). To mitigate this issue, *speculative decoding* (SD) introduces

| Methods | Drafting | | | Verification | | | Speedup |
|---------|----------|----|----|-------|--------|-----------|---------|
| | Approach | AM | Plug&Play | Greedy | Sampling | Token Tree | |
| EAGLE (Li et al., 2024a;b) | Draft Heads | Yes | ✗ | ✓ | ✓ | ✓ | - |
| REST (He et al., 2024) | Context Retrieval | Yes | ✗ | ✓ | ✓ | ✓ | - |
| SELF-SD (Zhang et al., 2024) | Layer Skipping | No | ✗ | ✓ | ✓ | ✗ | - |
| PARALLEL (Santilli et al., 2023) | Jacobi Decoding | No | ✓ | ✓ | ✗ | ✗ | 0.9×~1.0× |
| LOOKAHEAD (Fu et al., 2024) | Jacobi Decoding | No | ✓ | ✓ | ✓ | ✓ | 1.2×~1.4× |
| SWIFT (Ours) | Layer Skipping | No | ✓ | ✓ | ✓ | ✓ | 1.3×~1.6× |

Table 1: Comparison of SWIFT with existing SD methods. "*AM*" denotes whether the method requires auxiliary modules such as additional parameters or data stores. "*Greedy*", "*Sampling*", and "*Token Tree*" denote whether the method supports greedy decoding, multinomial sampling, and token tree verification, respectively. SWIFT is the first plug-and-play layer-skipping SD method, which is orthogonal to those Jacobi-based methods such as Lookahead (Fu et al., 2024).

utilizing a compact draft model to predict multiple decoding steps, with the target LLM then validating them in parallel (Xia et al., 2023; Leviathan et al., 2023; Chen et al., 2023). Recent SD variants have sought to enhance efficiency by incorporating additional modules (Kim et al., 2023; Sun et al., 2023; Du et al., 2024; Li et al., 2024a;b) or introducing new training objectives (Liu et al., 2023a; Kou et al., 2024; Zhou et al., 2024; Gloeckle et al., 2024). However, these approaches necessitate extra parameters or extensive training, limiting their applicability across different models. Another line of research has explored *plug-and-play* SD methods with Jacobi decoding (Santilli et al., 2023; Fu et al., 2024), which predict multiple steps in parallel by appending pseudo tokens to the input and refining them iteratively. As shown in Table 1, our work complements these efforts by investigating a novel plug-and-play SD method with *layer-skipping*, which exploits the inherent sparsity of LLM layers to accelerate inference. The most related approaches to ours include Self-SD (Zhang et al., 2024) and LayerSkip (Elhoushi et al., 2024), which also skip intermediate layers of LLMs to form the draft model. However, both methods require a time-consuming offline training process, making them neither plug-and-play nor easily generalizable across different models and tasks.

**Efficient LLMs Utilizing Sparsity** LLMs are powerful but often over-parameterized (Hu et al., 2022). To address this issue, various methods have been proposed to accelerate inference by leveraging different forms of LLM sparsity. One promising research direction is model compression, which includes approaches such as quantization (Dettmers et al., 2022; Frantar et al., 2023; Ma et al., 2024), parameter pruning (Liu et al., 2019; Hoefler et al., 2021; Liu et al., 2023b), and knowledge distillation (Touvron et al., 2021; Hsieh et al., 2023; Gu et al., 2024). These approaches aim to reduce model sparsity by compressing LLMs into more compact forms, thereby decreasing memory usage and computational overhead during inference. Our proposed method, SWIFT, focuses specifically on sparsity within LLM layer computations, providing a more streamlined approach to efficient LLM inference that builds upon recent advances in layer skipping (Corro et al., 2023; Zhu et al., 2024; Jaiswal et al., 2024; Liu et al., 2024). Unlike these existing layer-skipping methods that may lead to information loss and performance degradation, SWIFT investigates the utilization of layer sparsity to enable lossless acceleration of LLM inference.

## 3 PRELIMINARIES

### 3.1 SELF-SPECULATIVE DECODING

Unlike most SD methods that require additional parameters, self-speculative decoding (Self-SD) first proposed utilizing *parts* of an LLM as a compact draft model (Zhang et al., 2024). In each decoding step, this approach skips intermediate layers of the LLM to efficiently generate draft tokens; these tokens are then validated in parallel by the full-parameter LLM to ensure that the output distribution of the target LLM remains unchanged. The primary challenge of Self-SD lies in determining which layers, and how many, should be skipped – referred to as the *skipped layer set* – during the drafting stage, which is formulated as an optimization problem. Formally, given the input data $\mathcal{X}$ and the target LLM $\mathcal{M}_T$ with $L$ layers (including both attention and MLP layers), Self-SD aims to identify the optimal skipped layer set $\boldsymbol{z}$ that minimizes the average inference time per token:

$$\boldsymbol{z}^* = \arg\min_{\boldsymbol{z}} \frac{\sum_{\boldsymbol{x}\in\mathcal{X}} f\left(\boldsymbol{x} \mid \boldsymbol{z}; \boldsymbol{\theta}_{\mathcal{M}_T}\right)}{\sum_{\boldsymbol{x}\in\mathcal{X}} |\boldsymbol{x}|}, \quad \text{s.t. } \boldsymbol{z} \in \{0,1\}^L, \tag{1}$$

where $f(\cdot)$ is a black-box function that returns the inference latency of sample $x$, $z_i \in \{0, 1\}$ denotes whether layer $i$ of the target LLM is skipped when drafting, and $|x|$ represents the sample length. Self-SD addresses this problem through a Bayesian optimization process (Jones et al., 1998). Before inference, this process iteratively selects new inputs $z$ based on a Gaussian process (Rasmussen & Williams, 2006) and evaluates Eq (1) on the training set of $\mathcal{X}$. After a specified number of iterations, the best $z$ is considered an approximation of $z^*$ and is held fixed for inference.

While Self-SD has proven effective, its reliance on a time-intensive Bayesian optimization process poses certain limitations. For each task, Self-SD must sequentially evaluate all selected training samples during every iteration to optimize Eq (1); Moreover, the computational burden of Bayesian optimization escalates substantially with the number of iterations. As a result, processing just eight CNN/Daily Mail (Nallapati et al., 2016) samples for 1000 Bayesian iterations requires nearly 7.5 hours for LLaMA-2-13B and 20 hours for LLaMA-2-70B on an NVIDIA A6000 server. These computational demands restrict the generalizability of Self-SD across different models and tasks.

## 3.2 EXPERIMENTAL OBSERVATIONS

This subsection delves into Self-SD, exploring the *plug-and-play* potential of this layer-skipping SD paradigm for lossless LLM inference acceleration. Our key findings are detailed below.

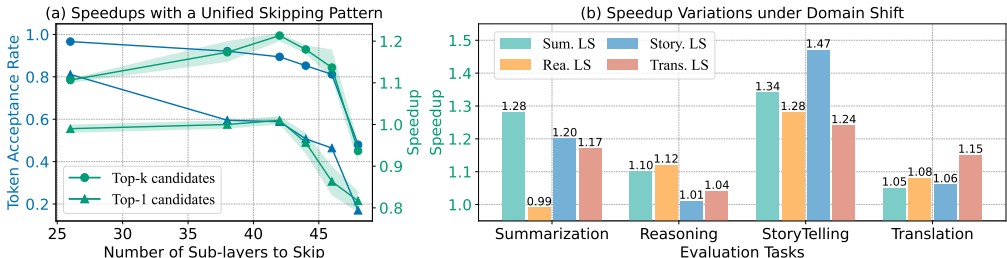

Figure 2: (a) LLMs possess self-acceleration potential via layer sparsity. By utilizing drafts from the top-$k$ candidates, we found that uniformly skipping half of the layers during drafting yields a notable $1.2\times$ speedup. (b) Layer sparsity is task-specific. Each task requires its own optimal set of skipped layers, and applying the skipped layer configuration from one task to another can lead to substantial performance degradation. "*X LS*" represents the skipped layer set optimized for task *X*.

### 3.2.1 LLMS POSSESS SELF-ACCELERATION POTENTIAL VIA LAYER SPARSITY

We begin by investigating the potential of behavior alignment between the target LLM and its *layer-skipping* variant. Unlike previous work (Zhang et al., 2024) that focused solely on greedy draft predictions, we leverage potential draft candidates from top-$k$ predictions, as detailed in Section 4.2. We conducted experiments using LLaMA-2-13B across the CNN/Daily Mail (Nallapati et al., 2016), GSM8K (Cobbe et al., 2021), and TinyStories (Eldan & Li, 2023) datasets. We applied a *uniform* layer-skipping pattern with $k$ set to 10. The experimental results, illustrated in Figure 2(a), demonstrate a $30\%$ average improvement in the token acceptance rate by leveraging top-$k$ predictions, with over $90\%$ of draft tokens accepted by the target LLM. Consequently, compared to Self-SD, which achieved a maximum speedup of $1.01\times$ in this experimental setting, we revealed that the *layer-skipping* SD paradigm could yield an average wall-clock speedup of $1.22\times$ over conventional autoregressive decoding *with a uniform layer-skipping pattern*. This finding challenges the prevailing belief that the selection of skipped layers must be meticulously curated, suggesting instead that LLMs possess greater potential for self-acceleration through inherent layer sparsity.

### 3.2.2 LAYER SPARSITY IS TASK-SPECIFIC

We further explore the following research question: *Is the skipped layer set optimized for one specific task applicable to other tasks?* To address this, we conducted domain shift experiments using LLaMA-2-13B on the CNN/Daily Mail, GSM8K, TinyStories, and WMT16 DE-EN datasets. The experimental results, depicted in Figure 2(b), reveal two critical findings: **1) Each task requires its own optimal skipped layer set.** As illustrated in Figure 2(b), the highest speedup performance is

consistently achieved by the skipped layer configuration specifically optimized for each task. The detailed configuration of these layers is presented in Appendix A, demonstrating that the optimal configurations differ across tasks. **2) Applying the static skipped layer configuration across different tasks can lead to substantial efficiency degradation.** For example, the speedup decreases from $1.47\times$ to $1.01\times$ when the optimized skipped layer set from a storytelling task is applied to a mathematical reasoning task, indicating that the optimized skipped layer set for one specific task does not generalize effectively to others.

These findings lay the groundwork for our *plug-and-play* solution within layer-skipping SD. Section 3.2.1 provides a strong foundation for real-time skipped layer selection, suggesting that additional optimization using training data may be unnecessary; Section 3.2.2 highlights the limitations of static layer-skipping patterns for dynamic input data streams across various tasks, underscoring the necessity for adaptive layer optimization during inference. Building on these insights, we present our *on-the-fly* self-speculative decoding method for efficient and adaptive layer set optimization.

# 4 SWIFT: ON-THE-FLY SELF-SPECULATIVE DECODING

We introduce SWIFT, the first *plug-and-play* self-speculative decoding approach that optimizes the skipped layer set of the target LLM on the fly, facilitating lossless LLM acceleration across diverse input data streams. As shown in Figure 3, SWIFT divides LLM inference into two distinct phases: (1) *context-based* layer set optimization (§4.1), which aims to identify the optimal skipped layer set given the input stream, and (2) *confidence-aware* inference acceleration (§4.2), which employs the determined configuration to accelerate LLM inference.

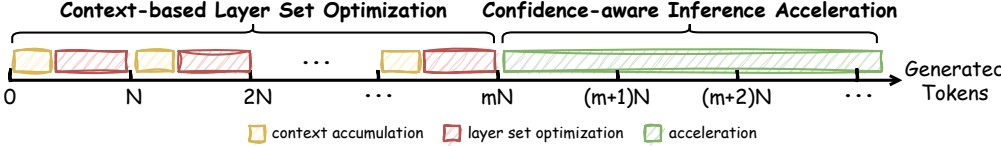

Figure 3: Timeline of SWIFT inference. *N* denotes the maximum generation length per instance.

## 4.1 CONTEXT-BASED LAYER SET OPTIMIZATION

Layer set optimization is a critical challenge in self-speculative decoding, as it determines which layers of the target LLM should be skipped to form the draft model (see Section 3.1). Unlike prior methods that rely on time-intensive offline optimization, our work emphasizes *on-the-fly* layer set optimization, which poses a greater challenge to the *latency-accuracy* trade-off: the optimization must be efficient enough to avoid delays during inference while ensuring accurate drafting of subsequent decoding steps. To address this, we propose an adaptive optimization mechanism that balances efficiency with drafting accuracy. Our method minimizes overhead by performing **only a single forward pass** of the draft model per step to validate potential skipped layer set candidates. The core innovation is the use of LLM-generated tokens (*i.e.*, prior context) as ground truth, allowing for simultaneous validation of the draft model's accuracy in predicting future decoding steps.

In the following subsections, we illustrate the detailed process of this optimization phase for *each input instance*, which includes context accumulation (§4.1.1) and layer set optimization (§4.1.2).

### 4.1.1 CONTEXT ACCUMULATION

Given an input instance in the optimization phase, the draft model is initialized by uniformly skipping layers in the target LLM. This initial layer-skipping pattern is maintained to accelerate inference until a specified number of LLM-generated tokens, referred to as the *context window*, has been accumulated. Upon reaching this window length, the inference transitions to layer set optimization.

### 4.1.2 LAYER SET OPTIMIZATION

During this stage, as illustrated in Figure 4, we integrate an optimization step before each LLM decoding step to refine the skipped layer set, which comprises two substeps:

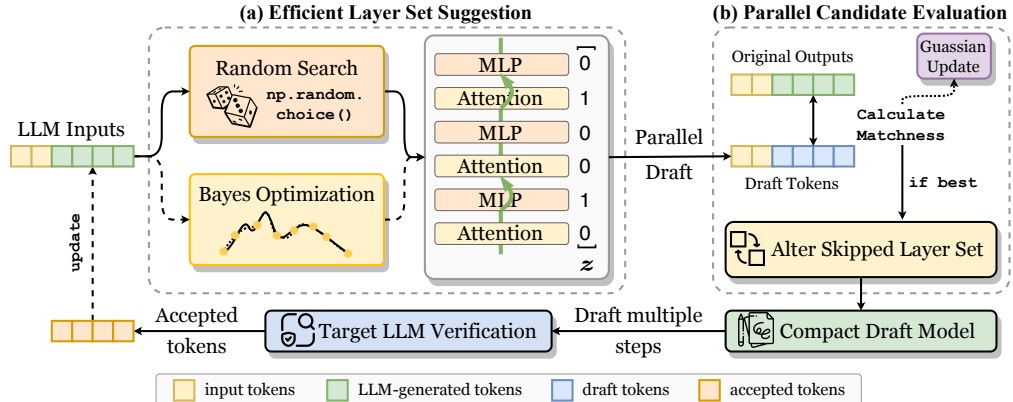

Figure 4: Layer set optimization process in SWIFT. During the optimization stage, SWIFT performs an optimization step prior to each LLM decoding step to adjust the skipped layer set, which involves: **(a) Efficient layer set optimization.** SWIFT integrates random search with interval Bayesian optimization to propose layer set candidates; **(b) Parallel candidate evaluation.** SWIFT uses LLM-generated tokens (*i.e.*, prior context) as ground truth, enabling simultaneous validation of the proposed candidates. The best-performing layer set is selected to accelerate the current decoding step.

**Efficient Layer Set Suggestion** This substep aims to suggest a potential layer set candidate. Formally, given a target LLM $\mathcal{M}_T$ with $L$ layers, our goal is to identify an optimal skipped layer set $\boldsymbol{z} \in \{0,1\}^L$ to form the compact draft model. Unlike Zhang et al. (2024), which relies entirely on a time-consuming Bayesian optimization process, we introduce an efficient strategy that combines random search with Bayesian optimization. In this approach, random sampling efficiently handles most of the exploration. Specifically, given a fixed skipping ratio $r$, SWIFT applies Bayesian optimization at regular intervals of $\beta$ optimization steps (*e.g.*, $\beta = 25$) to suggest the next layer set candidate, while random search is employed during other optimization steps.

$$\boldsymbol{z} = \begin{cases} \text{Bayesian\_Optimization}(\boldsymbol{l}) & \text{if } o \% \beta = 0 \\ \text{Random\_Search}(\boldsymbol{l}) & \text{otherwise} \end{cases}, \tag{2}$$

where $1 \leq o \leq S$ is the current optimization step; $S$ denotes the maximum number of optimization steps; $\boldsymbol{l} = \binom{L}{rL}$ denotes the input space, *i.e.*, all possible combinations of layers that can be skipped.

**Parallel Candidate Evaluation** SWIFT leverages LLM-generated context to simultaneously validate the candidate draft model's performance in predicting future decoding steps. Formally, given an input sequence $\boldsymbol{x}$ and the previously generated tokens within the context window, denoted as $\boldsymbol{y} = \{y_1, \ldots, y_\gamma\}$, the draft model $\mathcal{M}_D$, which skips the designated layers $\boldsymbol{z}$ of the target LLM, is employed to predict these context tokens in parallel:

$$y_i' = \arg\max_y \log P\left(y \mid \boldsymbol{x}, \boldsymbol{y}_{<i}; \boldsymbol{\theta}_{\mathcal{M}_D}\right), 1 \leq i \leq \gamma, \tag{3}$$

where $\gamma$ represents the context window. The cached key-value pairs in the target LLM $\mathcal{M}_T$ are reused by $\mathcal{M}_D$, presumably aligning $\mathcal{M}_D$'s distribution with $\mathcal{M}_T$ and reducing the redundant computation. The `matchness` score is defined as the exact match ratio between $\boldsymbol{y}$ and $\boldsymbol{y}'$:

$$\texttt{matchness} = \frac{\sum_i \mathbb{I}\left(y_i = y_i'\right)}{\gamma}, 1 \leq i \leq \gamma, \tag{4}$$

where $\mathbb{I}(\cdot)$ denotes the indicator function. This score serves as the *optimization objective* during optimization, reflecting $\mathcal{M}_D$'s accuracy in predicting future decoding steps. As shown in Figure 4, the `matchness` score at each step is integrated into the Gaussian process model to guide Bayesian optimization, with the highest-scoring layer set candidate being retained to form the draft model.

As illustrated in Figure 3, the process of context accumulation and layer set optimization alternates for each instance until a termination condition is met – either the maximum number of optimization steps is reached or the best candidate remains unchanged over multiple iterations. Once the optimization phase concludes, the inference process transitions to the confidence-aware inference acceleration phase, where the optimized draft model is employed to speed up LLM inference.

## 4.2 CONFIDENCE-AWARE INFERENCE ACCELERATION

During the acceleration phase, the optimization step is removed. SWIFT applies the best-performed layer set to form the compact draft model and decodes following the draft-then-verify paradigm. Specifically, at each decoding step, given the input $x$ and previous LLM outputs $y$, the draft model $\mathcal{M}_D$ predicts future LLM decoding steps in an autoregressive manner:

$$y'_j = \arg\max_y \log P\left(y \mid x, y, y'_{<j}; \boldsymbol{\theta}_{\mathcal{M}_D}\right), \quad (5)$$

where $1 \leq j \leq N_D$ is the current draft step, $N_D$ denotes the maximum draft length, $y'_{<j}$ represents previous draft tokens, and $P(\cdot)$ denotes the probability distribution of the next draft token. The KV cache of the target LLM $\mathcal{M}_T$ and preceding draft tokens $y'_{<j}$ is reused to reduce the computational cost.

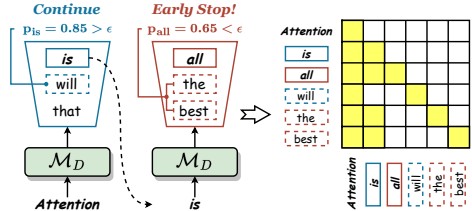

(a) Early-stopping Drafting     (b) Dynamic Verification

Figure 5: *Confidence-aware* inference process of SWIFT. (a) The drafting terminates early if the confidence score drops below threshold $\epsilon$. (b) Draft candidates are dynamically selected based on confidence and then verified in parallel by the target LLM.

Let $p_j = \max P(\cdot)$ denote the probability of the top-1 draft prediction $y'_j$, which can be regarded as a *confidence* score. Recent research (Li et al., 2024b; Du et al., 2024) shows that this score is highly correlated with the likelihood that the draft token $y'_j$ will pass verification – higher confidence scores indicate a greater chance of acceptance. Therefore, following previous studies (Zhang et al., 2024; Du et al., 2024), we leverage the confidence score to prune unnecessary draft steps and select valuable draft candidates, improving both speculation accuracy and verification efficiency.

As shown in Figure 5, we integrate SWIFT with two *confidence-aware* inference strategies[1]: **1) Early-stopping Drafting.** The autoregressive drafting process halts if the confidence $p_j$ falls below a specified threshold $\epsilon$, avoiding any waste of subsequent drafting computation. **2) Dynamic Verification.** Each $y'_j$ is dynamically extended with its top-$k$ draft predictions for parallel verification to enhance speculation accuracy, with $k$ determined by the confidence score $p_j$. Concretely, $k$ is set to 10, 5, 3, and 1 for $p$ in the ranges of $(0, 0.5]$, $(0.5, 0.8]$, $(0.8, 0.95]$, and $(0.95, 1]$, respectively. All draft candidates are linearized into a single sequence and verified in parallel by the target LLM using a special causal attention mask (see Figure 5 (b)).

# 5 EXPERIMENTS

## 5.1 EXPERIMENTAL SETUP

**Implementation Details**   We mainly evaluate SWIFT on LLaMA-2 (Touvron et al., 2023b) and CodeLLaMA series (Rozière et al., 2023) across various tasks, including summarization, mathematical reasoning, storytelling, and code generation. The evaluation datasets include CNN/Daily Mail (CNN/DM) (Nallapati et al., 2016), GSM8K (Cobbe et al., 2021), TinyStories (Eldan & Li, 2023), and HumanEval (Chen et al., 2021). The maximum generation lengths on CNN/DM, GSM8K, and TinyStories are set to 64, 64, and 128, respectively. We conduct 1-shot evaluation for CNN/DM and TinyStories, and 5-shot evaluation for GSM8K. We compare pass@1 and pass@10 for HumanEval. We randomly sample 1000 instances from the test set for each dataset except HumanEval. The maximum generation lengths for HumanEval and all analyses are set to 512. During optimization, we employ both random search and Bayesian optimization[2] to suggest skipped layer set candidates. Following prior work, we adopt speculative sampling (Leviathan et al., 2023) as our acceptance strategy with a batch size of 1. Detailed setups are provided in Appendix B.1 and B.2.

**Baselines**   In our main experiments, we compare SWIFT to two existing *plug-and-play* methods: Parallel Decoding (Santilli et al., 2023) and Lookahead Decoding (Fu et al., 2024), both of which employ Jacobi decoding for efficient LLM drafting. It is important to note that SWIFT, as a *layer-skipping* SD method, is orthogonal to these Jacobi-based SD methods, and integrating SWIFT with them could further boost inference efficiency. We exclude other SD methods from our comparison as they necessitate additional modules or extensive training, which limits their generalizability.

---

[1]These confidence-aware inference strategies are also applied during the optimization phase, where the current optimal layer set is used to form the draft model and accelerate the corresponding LLM decoding step.

[2]https://github.com/bayesian-optimization/BayesianOptimization

| Models | Methods | CNN/DM | | GSM8K | | TinyStories | | Speed (tokens/s) | Overall Speedup |
|---|---|---|---|---|---|---|---|---|---|
| | | $M$ | Speedup | $M$ | Speedup | $M$ | Speedup | | |
| LLaMA-2-13B | VANILLA | 1.00 | 1.00× | 1.00 | 1.00× | 1.00 | 1.00× | 20.10 | 1.00× |
| | PARALLEL | 1.04 | 0.95× | 1.11 | 0.99× | 1.06 | 0.97× | 19.49 | 0.97× |
| | LOOKAHEAD | 1.38 | 1.16× | 1.50 | 1.29× | 1.62 | 1.37× | 25.46 | 1.27× |
| | SWIFT | 4.34 | **1.37×**[†] | 3.13 | **1.31×**[†] | 8.21 | **1.53×**[†] | **28.26** | **1.41×** |
| LLaMA-2-13B -Chat | VANILLA | 1.00 | 1.00× | 1.00 | 1.00× | 1.00 | 1.00× | 19.96 | 1.00× |
| | PARALLEL | 1.06 | 0.96× | 1.08 | 0.97× | 1.10 | 0.98× | 19.26 | 0.97× |
| | LOOKAHEAD | 1.35 | 1.15× | 1.57 | **1.31×** | 1.66 | 1.40× | 25.69 | 1.29× |
| | SWIFT | 3.54 | **1.28×** | 2.95 | 1.25× | 7.42 | **1.50×**[†] | **26.80** | **1.34×** |
| LLaMA-2-70B | VANILLA | 1.00 | 1.00× | 1.00 | 1.00× | 1.00 | 1.00× | 4.32 | 1.00× |
| | PARALLEL | 1.05 | 0.95× | 1.07 | 0.97× | 1.05 | 0.96× | 4.14 | 0.96× |
| | LOOKAHEAD | 1.36 | 1.15× | 1.54 | 1.30× | 1.59 | 1.35× | 5.45 | 1.26× |
| | SWIFT | 3.85 | **1.43×**[†] | 2.99 | **1.39×**[†] | 6.17 | **1.62×**[†] | **6.41** | **1.48×** |

Table 2: Comparison between SWIFT and prior plug-and-play methods. We report the mean generated length $M$, speedup ratio, and average decoding speed (tokens/s) under greedy decoding. [†] indicates results with a token acceptance rate $\alpha$ above 0.98. More details are provided in Appendix C.1.

| Datasets | Methods | CodeLLaMA-13B | | | | CodeLLaMA-34B | | | |
|---|---|---|---|---|---|---|---|---|---|
| | | $M$ | $\alpha$ | Acc. | Speedup | $M$ | $\alpha$ | Acc. | Speedup |
| HumanEval (pass@1) | VANILLA | 1.00 | - | 0.311 | 1.00× | 1.00 | - | 0.372 | 1.00× |
| | SWIFT | 4.75 | 0.98 | 0.311 | **1.40×** | 3.79 | 0.88 | 0.372 | **1.46×** |
| HumanEval (pass@10) | VANILLA | 1.00 | - | 0.628 | 1.00× | 1.00 | - | 0.677 | 1.00× |
| | SWIFT | 3.55 | 0.93 | 0.628 | **1.29×** | 2.79 | 0.90 | 0.683 | **1.30×** |

Table 3: Experimental results of SWIFT on code generation tasks. We report the mean generated length $M$, acceptance rate $\alpha$, accuracy (Acc.), and speedup ratio for comparison. We use greedy decoding for pass@1 and random sampling with a temperature of 0.6 for pass@10.

**Evaluation Metrics** We report two widely-used metrics for SWIFT evaluation: mean generated length $M$ (Stern et al., 2018) and token acceptance rate $\alpha$ (Leviathan et al., 2023). Detailed descriptions of these metrics can be found in Appendix B.3. In addition to these metrics, we report the actual decoding speed (tokens/s) and wall-time speedup ratio compared with vanilla autoregressive decoding. The acceleration of SWIFT theoretically guarantees the preservation of the target LLMs' output distribution, making it unnecessary to evaluate the generation quality. However, to provide a point of reference, we present the evaluation scores for code generation tasks.

## 5.2 MAIN RESULTS

Table 2 presents the comparison between SWIFT and previous plug-and-play methods on text generation tasks. The experimental results demonstrate the following findings: (1) SWIFT shows superior efficiency over prior methods, achieving consistent speedups of 1.3×∼1.6× over vanilla autoregressive decoding across various models and tasks. (2) The efficiency of SWIFT is driven by the high behavior consistency between the target LLM and its layer-skipping draft variant. As shown in Table 2, SWIFT produces a mean generated length $M$ of 5.01, with a high token acceptance rate $\alpha$ ranging from 90% to 100%. Notably, for the LLaMA-2 series, this acceptance rate remains stable at 98%∼100%, indicating that nearly all draft tokens are accepted by the target LLM. (3) Compared with 13B models, LLaMA-2-70B achieves higher speedups with a larger layer skip ratio (0.45→0.5), suggesting that larger-scale LLMs exhibit greater layer sparsity. This underscores SWIFT's potential to deliver even greater speedups as LLM scales continue to grow. A detailed analysis of this finding is presented in Section 5.3, while additional experimental results for LLaMA-70B models, including LLaMA-3-70B, are presented in Appendix C.2.

Table 3 shows the evaluation results of SWIFT on code generation tasks. SWIFT achieves speedups of 1.3×∼1.5× over vanilla autoregressive decoding, demonstrating its effectiveness across both greedy decoding and random sampling settings. Additionally, speculative sampling theoretically guarantees that SWIFT maintains the original output distribution of the target LLM. This is empirically validated

by the task performance metrics in Table 3. Despite a slight variation in the pass@10 metric for CodeLLaMA-34B, SWIFT achieves identical performance to autoregressive decoding.

## 5.3 IN-DEPTH ANALYSIS

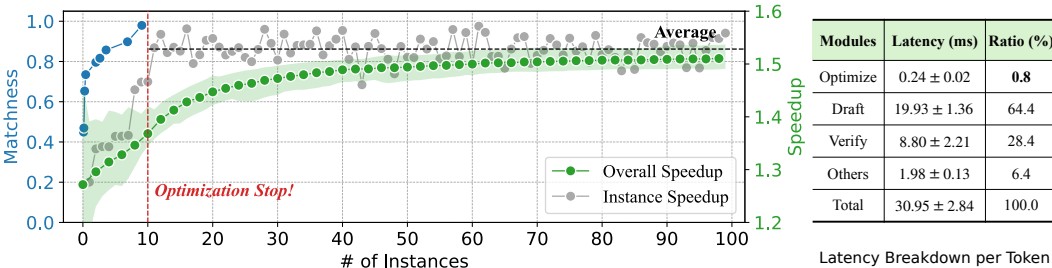

| Modules | Latency (ms) | Ratio (%) |
|---|---|---|
| Optimize | $0.24 \pm 0.02$ | **0.8** |
| Draft | $19.93 \pm 1.36$ | 64.4 |
| Verify | $8.80 \pm 2.21$ | 28.4 |
| Others | $1.98 \pm 0.13$ | 6.4 |
| Total | $30.95 \pm 2.84$ | 100.0 |

Latency Breakdown per Token

Figure 6: Illustration and latency breakdown of SWIFT inference. As the left figure shows, after the context-based layer set optimization phase, the overall speedup of SWIFT steadily increases, reaching the average instance speedup during the acceleration phase. The additional optimization steps account for only **0.8**% of the total inference latency, as illustrated in the right figure.

**Illustration of Inference** As described in Section 4, SWIFT divides the LLM inference process into two distinct phases: *optimization* and *acceleration*. Figure 6 (*left*) illustrates the detailed acceleration effect of SWIFT during LLM inference. Specifically, the **optimization** phase begins at the start of inference, where an optimization step is performed before each decoding step to adjust the skipped layer set forming the draft model. As shown in Figure 6, in this phase, the `matchness` score of the draft model rises sharply from 0.45 to 0.73 during the inference of the first instance. This score then gradually increases to 0.98, which triggers the termination of the optimization process. Subsequently, the inference transitions to the **acceleration** phase, during which the optimization step is removed, and the draft model remains fixed to accelerate LLM inference. As illustrated, the instance speedup increases with the `matchness` score, reaching an average of $1.53\times$ in the acceleration phase. The overall speedup gradually rises as more tokens are generated, eventually approaching the average instance speedup. This dynamic reflects a key feature of SWIFT: **the efficiency of SWIFT improves with increasing input length and the number of instances**.

**Breakdown of Computation** Figure 6 (*right*) presents the computation breakdown of different modules in SWIFT with 1000 CNN/DM samples using LLaMA-2-13B. The results demonstrate that the optimization step only takes **0.8**% of the overall inference process, indicating the efficiency of our strategy. Compared with Self-SD (Zhang et al., 2024) that requires a time-consuming optimization process (*e.g.*, 7.5 hours for LLaMA-2-13B on CNN/DM), SWIFT achieves a nearly **180**× optimization time reduction, facilitating *on-the-fly* inference acceleration. Besides, the results show that the drafting stage of SWIFT consumes the majority of inference latency. This is consistent with our results of *mean generated length* in Table 2 and 3, which shows that nearly 80% output tokens are generated by the efficient draft model, demonstrating the effectiveness of our SWIFT framework.

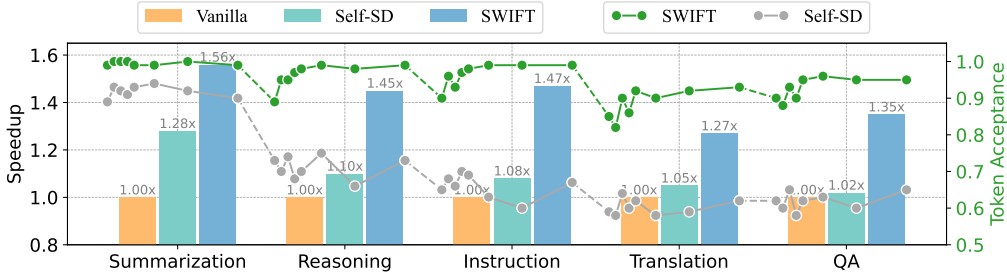

Figure 7: Comparison between SWIFT and Self-SD in handling dynamic data input streams. Unlike Self-SD, which suffers from efficiency reduction during distribution shift, SWIFT maintains stable acceleration performance with an acceptance rate exceeding 0.9.

**Dynamic Input Data Streams** We further validate the effectiveness of SWIFT in handling dynamic input data streams. We selected CNN/DM, GSM8K, Alpaca (Taori et al., 2023), WMT14 DE-EN, and Nature Questions (Kwiatkowski et al., 2019) for the evaluation on summarization, reasoning, instruction following, translation, and question answering tasks, respectively. For each task, we randomly sample 500 instances from the test set and concatenate them task-by-task to form the input stream. The experimental results are presented in Figure 7. As demonstrated, Self-SD is sensitive to domain shifts, with the average token acceptance rate dropping from 92% to 68%. Consequently, it suffers from severe speedup reduction from $1.33\times$ to an average of $1.05\times$ under domain shifts. In contrast, SWIFT exhibits promising adaptation capability to different domains with an average token acceptance rate of 96%, leading to a consistent $1.3\times\sim1.6\times$ speedup.

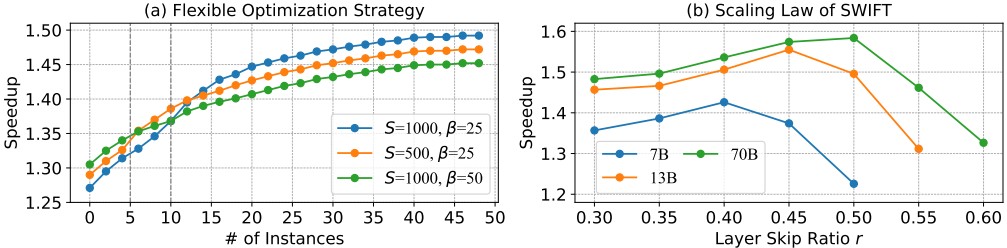

Figure 8: In-depth analysis of SWIFT, which includes: **(a) Flexible optimization strategy.** The maximum optimization iteration $S$ and Bayesian interval $\beta$ can be flexibly adjusted to accommodate different input data types. **(b) Scaling law.** The speedup and optimal layer skip ratio of SWIFT increase with larger model sizes, indicating that larger LLMs exhibit greater layer sparsity.

**Flexible Optimization & Scaling Law** Figure 8(a) presents the flexibility of SWIFT in handling various input types by adjusting the maximum optimization step $S$ and Bayesian interval $\beta$. For input with fewer instances, reducing $S$ enables an earlier transition to the acceleration phase while increasing $\beta$ reduces the overhead during the optimization phase, enhancing speedups during the initial stages of inference. In cases with sufficient input data, SWIFT enables exploring more optimization paths, thereby enhancing the overall speedup. Figure 8(b) illustrates the scaling law of SWIFT: as the model size increases, both the optimal layer-skip ratio and overall speedup improve, indicating that larger LLMs exhibit more layer sparsity. This finding highlights the potential of SWIFT for accelerating LLMs of larger sizes (*e.g.*, 175B), which we leave for future investigation.

**Other LLM Backbones** Beyond LLaMA, we assess the effectiveness of SWIFT on additional LLM backbones. Specifically, we include Yi-34B (Young et al., 2024) and DeepSeek-Coder-33B (Guo et al., 2024) along with their instruction-tuned variants for text and code generation tasks, respectively. The speedup results of SWIFT are illustrated in Figure 9, demonstrating that SWIFT achieves efficiency improvements ranging from 26% to 54% on these LLM backbones. Further experimental details are provided in Appendix C.3.

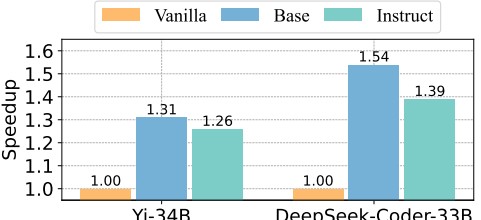

Figure 9: Speedups of SWIFT on LLM backbones and their instruction-tuned variants.

## 6 CONCLUSION

In this work, we introduce SWIFT, an on-the-fly self-speculative decoding algorithm that adaptively selects certain intermediate layers of LLMs to skip during inference. The proposed method does not require additional training or auxiliary models, making it a *plug-and-play* solution for accelerating LLM inference across diverse input data streams. Extensive experiments conducted across various LLMs and tasks demonstrate that SWIFT achieves over a $1.3\times\sim1.6\times$ speedup while preserving the distribution of the generated text. Furthermore, our in-depth analysis highlights the effectiveness of SWIFT in handling dynamic input data streams and its seamless integration with various LLM backbones, showcasing the great potential of this paradigm for practical LLM inference acceleration.

## ETHICS STATEMENT

The datasets used in our experiments are publicly released and labeled through interaction with humans in English. In this process, user privacy is protected, and no personal information is contained in the dataset. The scientific artifacts that we used are available for research with permissive licenses. The use of these artifacts in this paper is consistent with their intended purpose.

## ACKNOWLEDGEMENTS

We thank all anonymous reviewers for their valuable comments during the review process. The work described in this paper was supported by Research Grants Council of Hong Kong (PolyU/15207122, PolyU/15209724, PolyU/15207821, PolyU/15213323) and PolyU internal grants (BDWP).

## REPRODUCIBILITY STATEMENT

All the results in this work are reproducible. We provide all the necessary code in the Supplementary Material to replicate our results. The repository includes environment configurations, scripts, and other relevant materials. We discuss the experimental settings in Section 5.1 and Appendix C, including implementation details such as models, datasets, inference setup, and evaluation metrics.

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

APPENDIX

## A    PRELIMINARY DETAILS

We present the detailed configuration of Self-SD across four task domains in Figure 10, demonstrating that the optimal skipped layer configurations vary depending on the specific task.

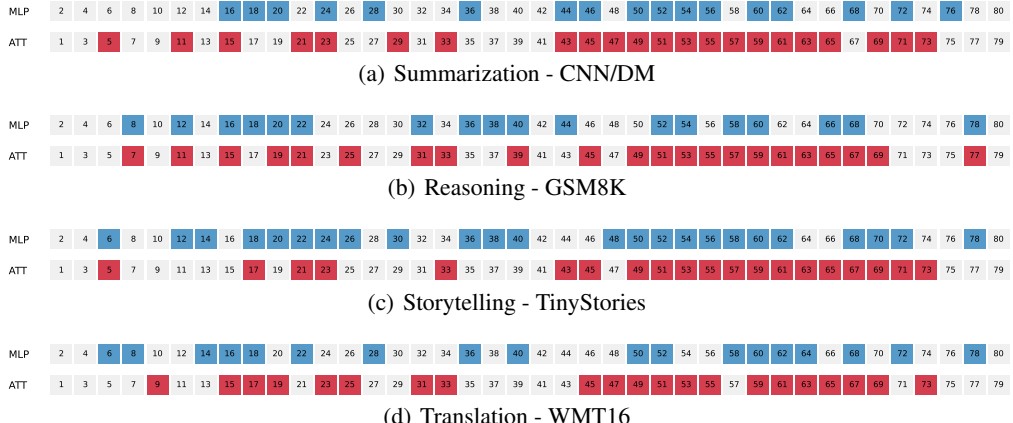

(a) Summarization - CNN/DM

(b) Reasoning - GSM8K

(c) Storytelling - TinyStories

(d) Translation - WMT16

Figure 10: Visualization of skipped layer set configurations of LLaMA-2-13B optimized by Self-SD (Zhang et al., 2024) on different task domains. Gray squares indicate retained layers, red squares denote skipped attention layers, and blue squares signify skipped MLP layers.

## B    EXPERIMENTAL SETUPS

### B.1    MODELS AND DATASETS

Our experiments mainly evaluate the effectiveness of SWIFT on LLaMA-2 (Touvron et al., 2023b) and CodeLLaMA series (Rozière et al., 2023). We provide empirical validation on a diverse range of generation tasks. For summarization, mathematical reasoning, storytelling, and code generation tasks, we chose the CNN/Daily Mail (CNN/DM) (Nallapati et al., 2016), GSM8K (Cobbe et al., 2021), TinyStories (Eldan & Li, 2023), and HumanEval (Chen et al., 2021) datasets, respectively. We perform 1-shot evaluation for CNN/DM and TinyStories, and 5-shot evaluation for GSM8K. The maximum generation lengths on CNN/DM, GSM8K, and TinyStories are set to 64, 64, and 128, respectively. We compare pass@1 and pass@10 for HumanEval. In our further analysis, we include three more datasets to validate the capability of SWIFT in handling dynamic input data streams. Specifically, we select Alpaca (Taori et al., 2023), WMT14 DE-EN, and Nature Questions (Kwiatkowski et al., 2019) for the instruction following, translation, and question answering tasks, respectively. The maximum generation lengths for HumanEval and all analyses are set to 512. We randomly sample 1000 instances from the test set for each dataset except HumanEval.

### B.2    INFERENCE SETUP

In the optimization phase, we employ both random search and Bayesian optimization to suggest potential skipped layer set candidates, striking a balance between optimization performance and efficiency. The context window $\gamma$ is set to 32. The maximum draft length $N_D$ is set to 25. For random sampling in code generation tasks, we apply a temperature of 0.6 and $top\_p = 0.95$. The maximum number of layer set optimization steps $S$ is set to 1000, with Bayesian optimization performed every $\beta = 25$ steps. The optimization phase is set to be early stopped if the `matchness` score does not improve after 300 steps or exceeds 0.95. The layer skip ratio $r$ is fixed at 0.45 for the 13B model and 0.5 for the 34B and 70B models. All experiments were conducted using Pytorch 2.1.0 on 4×NVIDIA RTX A6000 GPU (40GB) with CUDA 12.1, and an Intel(R) Xeon(R) Platinum 8370C

CPU with 32 cores. Inference for our method and all baselines was performed using the Hugging-face transformers package. Following prior work, we adopt speculative sampling (Leviathan et al., 2023) as our acceptance strategy, and the batch size is set to 1.

### B.3 EVALUATION METRICS

This subsection provides a detailed illustration of our evaluation metrics, which are mean generated length $M$ and token acceptance rate $\alpha$. Specifically, the mean generated length $M$ refers to the average number of output tokens produced per forward pass of the target LLM; the acceptance rate $\alpha$ is defined as the ratio of accepted tokens to the total number of draft steps. In other words, it represents the expected probability of the target LLM accepting a potential token from a forward pass of the draft model. These two metrics are independent of computational hardware and, therefore considered as more objective metrics. Given the mean generated length $M$, acceptance rate $\alpha$, and the layer skip ratio $r$, the mathematical formula for the expected wall-time speedup during the acceleration phase is derived as follows:

$$\mathbb{E}(\text{Spd.}) = \frac{M}{(M-1) \times \frac{c}{\alpha} + 1} = \frac{M\alpha}{(M-1)c + \alpha}, \quad c = 1 - r, \tag{6}$$

where $c$ is defined as the *cost coefficient* in Leviathan et al. (2023). It is calculated as the ratio between the single forward time of the draft model and that of the target LLM. In summary, the ideal speedup will be higher with the larger $M$ and $\alpha$ and smaller $c$.

## C EXPERIMENTAL DETAILS

### C.1 DETAILS OF MAIN RESULTS

We present the detailed statistics of our main experimental results in Table 4. SWIFT consistently achieves a token acceptance rate $\alpha$ exceeding $90\%$ across all evaluation settings, with the mean generated length $M$ ranging from 2.99 to 8.21. These statistics indicate strong behavior alignment between the target LLM and its layer-skipping draft variant, as discussed in Section 5.2. Additionally, we report the expected speedup $\mathbb{E}(\text{Spd.})$ calculated using Eq (6), indicating that the current implementation of SWIFT has significant potential for further optimization to boost its efficiency.

| Models | Methods | CNN/DM | | | GSM8K | | | TinyStories | | | Expected Speedup |
|---|---|---|---|---|---|---|---|---|---|---|---|
| | | $M$ | $\alpha$ | $\mathbb{E}(\text{Spd.})$ | $M$ | $\alpha$ | $\mathbb{E}(\text{Spd.})$ | $M$ | $\alpha$ | $\mathbb{E}(\text{Spd.})$ | |
| LLaMA-2-13B | VANILLA | 1.00 | - | 1.00× | 1.00 | - | 1.00× | 1.00 | - | 1.00× | 1.00× |
| | SWIFT | 4.34 | 0.99 | **1.52×** | 3.13 | 0.98 | **1.43×** | 8.21 | 1.00 | **1.65×** | **1.53×** |
| LLaMA-2-13B -Chat | VANILLA | 1.00 | - | 1.00× | 1.00 | - | 1.00× | 1.00 | - | 1.00× | 1.00× |
| | SWIFT | 3.54 | 0.90 | **1.39×** | 2.95 | 0.92 | **1.36×** | 7.42 | 0.99 | **1.62×** | **1.46×** |
| LLaMA-2-70B | VANILLA | 1.00 | - | 1.00× | 1.00 | - | 1.00× | 1.00 | - | 1.00× | 1.00× |
| | SWIFT | 3.85 | 0.99 | **1.58×** | 2.99 | 0.98 | **1.48×** | 6.17 | 0.99 | **1.71×** | **1.59×** |

Table 4: Detailed results of SWIFT on text generation tasks using LLaMA-2 series. We report the mean generated length $M$, token acceptance rate $\alpha$, and the expected speedup $\mathbb{E}(\text{Spd.})$ calculated by Eq (6) under the setting of greedy decoding with FP16 precision.

### C.2 ADDITIONAL RESULTS ON LLAMA-70B MODELS

In addition to the main results presented in Table 2, we provide further experimental evaluations of SWIFT on LLaMA-70B models, including LLaMA-2-70B and LLaMA-3-70B, along with their instruction-tuned variants, under the same experimental settings. The results demonstrate that SWIFT consistently achieves a $1.4\times\sim1.5\times$ wall-clock speedup across both the LLaMA-2 and LLaMA-3 series. Notably, SWIFT achieves a token acceptance rate $\alpha$ exceeding $85\%$ across various evaluation settings, with the mean generated length M ranging from 3.43 to 7.80. Although differences in layer redundancy are observed between models (e.g., skip ratio $r$ for LLaMA-2-70B

vs. LLaMA-3-70B[3]), SWIFT demonstrates robust adaptability, maintaining consistent acceleration performance regardless of model version.

| Models | Methods | CNN/DM | | | GSM8K | | | TinyStories | | | Overall Speedup |
|--------|---------|--------|---|---------|-------|---|---------|-------------|---|---------|-----------------|
| | | $M$ | $\alpha$ | Speedup | $M$ | $\alpha$ | Speedup | $M$ | $\alpha$ | Speedup | |
| LLaMA-2-70B | VANILLA | 1.00 | - | 1.00× | 1.00 | - | 1.00× | 1.00 | - | 1.00× | 1.00× |
| | SWIFT | 3.85 | 0.99 | **1.43×** | 2.99 | 0.98 | **1.39×** | 6.17 | 0.99 | **1.62×** | **1.48×** |
| LLaMA-2-70B -Chat | VANILLA | 1.00 | - | 1.00× | 1.00 | - | 1.00× | 1.00 | - | 1.00× | 1.00× |
| | SWIFT | 3.43 | 0.85 | **1.31×** | 3.12 | 0.89 | **1.32×** | 5.45 | 0.95 | **1.53×** | **1.37×** |
| LLaMA-3-70B | VANILLA | 1.00 | - | 1.00× | 1.00 | - | 1.00× | 1.00 | - | 1.00× | 1.00× |
| | SWIFT | 5.43 | 0.99 | **1.41×** | 4.11 | 0.99 | **1.37×** | 7.80 | 0.99 | **1.51×** | **1.43×** |
| LLaMA-3-70B -Instruct | VANILLA | 1.00 | - | 1.00× | 1.00 | - | 1.00× | 1.00 | - | 1.00× | 1.00× |
| | SWIFT | 3.76 | 0.95 | **1.33×** | 3.92 | 0.93 | **1.31×** | 5.87 | 0.97 | **1.43×** | **1.36×** |

Table 5: Experimental results of SWIFT on text generation tasks using the LLaMA-70B series. We report the mean generated length $M$, token acceptance rate $\alpha$, and speedup ratio under the setting of greedy decoding. The skip ratio $r$ is set to 0.5 for LLaMA-2 models and 0.4 for LLaMA-3 models.

## C.3 DETAILED RESULTS OF LLM BACKBONES

To further validate the effectiveness of SWIFT, we conducted experiments using additional LLM backbones beyond the LLaMA series. Specifically, we select two recently representative LLMs: Yi-34B for text generation and DeepSeek-Coder-33B for code generation tasks. The experimental results are illustrated in Table 6 and 7, demonstrating the efficacy of SWIFT across these LLM backbones. SWIFT achieves a consistent 1.2×∼1.3× wall-clock speedup on the Yi-34B series and a 1.3×∼1.5× on the DeepSeek-Coder-33B series. Notably, for the DeepSeek-Coder-33B series, SWIFT attains a mean generate length $M$ ranging from 3.16 to 4.17, alongside a token acceptance rate $\alpha$ exceeding 83%. These findings substantiate the utility of SWIFT as a general-purpose, *plug-and-play* SD method, offering promising inference acceleration across diverse LLM backbones.

| Models | Methods | CNN/DM | | | GSM8K | | | TinyStories | | | Overall Speedup |
|--------|---------|--------|---|---------|-------|---|---------|-------------|---|---------|-----------------|
| | | $M$ | $\alpha$ | Speedup | $M$ | $\alpha$ | Speedup | $M$ | $\alpha$ | Speedup | |
| Yi-34B | VANILLA | 1.00 | - | 1.00× | 1.00 | - | 1.00× | 1.00 | - | 1.00× | 1.00× |
| | SWIFT | 2.74 | 0.94 | **1.30×** | 2.65 | 0.97 | **1.28×** | 3.25 | 0.98 | **1.34×** | **1.31×** |
| Yi-34B-Chat | VANILLA | 1.00 | - | 1.00× | 1.00 | - | 1.00× | 1.00 | - | 1.00× | 1.00× |
| | SWIFT | 2.84 | 0.91 | **1.29×** | 2.77 | 0.89 | **1.27×** | 2.52 | 0.80 | **1.21×** | **1.26×** |

Table 6: Experimental results of SWIFT on text generation tasks using Yi-34B series. We report the mean generated length $M$, token acceptance rate $\alpha$ and speedup ratio under the setting of greedy decoding with FP16 precision. The skip ratio $r$ is set to 0.45.

## D FURTHER ANALYSIS AND DISCUSSION

### D.1 ABLATION STUDY

Table 8 presents the ablation study of SWIFT using LLaMA-2-13B on CNN/DM. The experimental results demonstrate that each component of SWIFT contributes to its overall speedup of SWIFT. Specifically, *early-stopping drafting* effectively reduces the number of ineffective draft steps, improving the token acceptance rate $\alpha$ by 55%. *Dynamic verification* further enhances efficiency by selecting suitable draft candidates from the top-$k$ predictions based on their confidence scores; removing this component leads to a decrease in both the mean generated length ($M$) and the overall speedup ratio. Additionally, the *optimization* phase refines the set of skipped layers, improving

---

[3]During the optimization phase, the layer skip ratio $r$ for LLaMA-3-70B was automatically adjusted from 0.5 to 0.4 as the token acceptance rate $\alpha$ remained below the tolerance threshold of 0.7.

speedup by $34\%$ compared to the initial uniform layer-skipping strategy. In summary, these results confirm the effectiveness of each proposed innovation in SWIFT.

| Datasets | Methods | DS-Coder | | | DS-Coder-Instruct | | |
|---|---|---|---|---|---|---|---|
| | | $M$ | $\alpha$ | Speedup | $M$ | $\alpha$ | Speedup |
| HumanEval (pass@1) | VANILLA | 1.00 | - | $1.00\times$ | 1.00 | - | $1.00\times$ |
| | SWIFT | 4.97 | 0.99 | $\mathbf{1.54\times}$ | 3.80 | 0.88 | $\mathbf{1.39\times}$ |
| HumanEval (pass@10) | VANILLA | 1.00 | - | $1.00\times$ | 1.00 | - | $1.00\times$ |
| | SWIFT | 3.16 | 0.91 | $\mathbf{1.36\times}$ | 3.74 | 0.83 | $\mathbf{1.31\times}$ |

Table 7: Experimental results of SWIFT on code generation tasks using DeepSeek-Coder-33B series. The skip ratio $r$ is set to 0.5. We use greedy decoding for pass@1 and random sampling with a temperature of 0.6 for pass@10. "*DS*" denotes the abbreviation of *DeepSeek*.

| Methods | $M$ | $\alpha$ | Speedup |
|---|---|---|---|
| Vanilla | 1.0 | - | $1.000\times$ |
| SWIFT | 5.82 | 0.98 | $\mathbf{1.560\times}$ |
| *w/o early-stopping* | 11.16 | 0.43 | $0.896\times$ |
| *w/o dynamic ver.* | 4.39 | 0.90 | $1.342\times$ |
| *w/o optimization* | 2.15 | 0.90 | $1.224\times$ |

Table 8: Ablation study of SWIFT. "*ver.*" denotes the abbreviation of *verification*.

| $\gamma$ | $M$ | $\alpha$ | Speedup | Latency |
|---|---|---|---|---|
| 16 | 3.91 | 0.95 | $1.341\times$ | 0.242ms |
| 32 | **5.82** | **0.98** | $\mathbf{1.560\times}$ | 0.244ms |
| 64 | 5.56 | 0.99 | $1.552\times$ | 0.312ms |
| 128 | 5.61 | 0.98 | $1.550\times$ | 0.425ms |

Table 9: Speedups of SWIFT across different context window $\gamma$. The *latency* of the optimization step is reported to illustrate the associated overhead.

## D.2 CONTEXT WINDOW

In Table 9, we show a detailed analysis of context window $\gamma$, which determines the number of LLM-generated tokens used in the layer set optimization process. A smaller $\gamma$ introduces greater randomness in the `matchness` score calculation, resulting in suboptimal performance, while a larger $\gamma$ increases the computational overhead of the optimization step. The results indicate that $\gamma = 32$ provides an optimal balance between optimization performance and computational overhead.

## D.3 COMPARISONS WITH PRIOR LAYER-SKIPPING METHODS

In this subsection, we compare SWIFT with two representative layer-skipping speculative decoding (SD) methods: LayerSkip (Elhoushi et al., 2024) and Self-SD (Zhang et al., 2024). Specifically, LayerSkip (Elhoushi et al., 2024) introduces an innovative approach to self-speculative decoding by implementing early-exit drafting, where the LLM generates drafts using only its earlier layers. However, this method necessitates a time-consuming pretraining or finetuning process, which modifies the original output distribution of the target LLM. Such alterations may compromise the reliability of the generated outputs; Self-SD (Zhang et al., 2024) proposed to construct the compact draft model by skipping intermediate layers, using an extensive Bayesian Optimization process before inference to determine the optimal skipped layers within the target LLM. As illustrated in Section 3.1, while effective, Self-SD suffers from significant optimization latency (nearly 7.5 hours for LLaMA-2-13B and 20 hours for LLaMA-2-70B). This prolonged optimization process limits its practicality and generalizability across diverse models and tasks.

Tables 10 and 11 summarize the comparative results in terms of acceleration performance and training/optimization costs, respectively. Below, we detail the advantages of SWIFT over these methods:

- **Comparison with LayerSkip:** LayerSkip achieves an aggressive skip ratio ($r = 0.8$), resulting in an average generated length of $2.42$ and a token acceptance rate of $0.64$. However, its reliance on pretraining or finetuning alters the ***original distribution*** of the target LLM, potentially reducing reliability. In contrast, SWIFT preserves the original distribution of the target LLM while delivering a comparable $1.56\times$ speedup without requiring additional training.

| Methods | Plug&Play | Original | $r$ | $M$ | $\alpha$ | Speedup |
|---|---|---|---|---|---|---|
| LAYERSKIP | ✗ | ✗ | 0.80 | 2.42 | 0.64 | 1.64× |
| SELF-SD | ✗ | ✓ | 0.43 | 4.02 | 0.85 | 1.29× |
| SELF-SD *w/ dynamic ver.* | ✗ | ✓ | 0.43 | 5.69 | 0.98 | 1.52× |
| SWIFT (Ours) | ✓ | ✓ | 0.45 | 5.82 | 0.98 | **1.56×** |

Table 10: Comparison of SWIFT and prior layer-skipping SD methods. We report the skip ratio $r$, mean generated length $M$, token acceptance $\alpha$, and speedup ratio under greedy decoding. The results are obtained with LLaMA-2-13B on CNN/DM. "*ver.*" denotes the abbreviation of *verification*.

| Methods | Training Cost | Optimization Latency |
|---|---|---|
| LAYERSKIP | $50 \times 10^3$ training steps with 64 A100 (80GB) | - |
| SELF-SD | 1000 Bayesian Optimization Iterations Before inference | ∼7.5 hours |
| SWIFT (Ours) | **N/A** | **∼2 minutes** |

Table 11: Comparison of SWIFT and prior layer-skipping SD methods in terms of training cost and optimization latency for LLaMA-2-13B. Training costs are sourced from the original papers, while optimization latency is measured from our re-implementation on an A6000 GPU. SWIFT demonstrates a ∼**200×** reduction in optimization latency compared to previous methods without requiring additional training, establishing it as an efficient plug-and-play SD method.

- **Comparison with Self-SD:** Self-SD relies on a time-intensive Bayesian Optimization process, which incurs substantial latency before inference. SWIFT eliminates this bottleneck through an on-the-fly optimization strategy, achieving an approximately **200×** reduction in optimization latency while maintaining the same 1.56× speedup. We further augmented Self-SD with our *Confidence-aware Inference Acceleration* strategy (Self-SD *w/ dynamic ver.*). Even compared to this augmented version, SWIFT achieves competitive speedups.

These findings highlight the efficiency and practicality of SWIFT over previous layer-skipping SD methods. As the first plug-and-play layer-skipping SD approach, we hope that SWIFT could provide valuable insights and inspire further research in this area.

### D.4 DETAILED COMPARISONS WITH SELF-SD

In this subsection, we provide a detailed comparison of SWIFT and Self-SD (Zhang et al., 2024). Figure 11 presents the speedups of Self-SD across varying optimization latencies, reflecting the increase in Bayesian Optimization iterations. As shown, Self-SD achieves minimal speedup improvement – almost equivalent to unified skipping – with fewer than 50 Bayesian iterations, corresponding to an optimization latency below 1474 seconds. At 100 Bayesian iterations, Self-SD achieves a 1.19× speedup; however, its optimization latency is nearly 25 times longer than that of SWIFT (2898s vs. 116s).

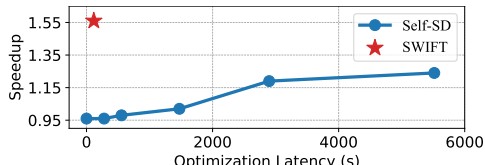

Figure 11: Comparison of SWIFT and Self-SD in terms of optimization latency and speedup. SWIFT achieves a 1.56× speedup with an optimization latency of 116 seconds.

Table 12 compares SWIFT and Self-SD (first two rows) under similar optimization latencies. The results highlight SWIFT's superiority in both optimization efficiency (116s vs. 155s) and speedup (1.56× vs. 0.97×). Even when compared to the augmented version of Self-SD (*w/ dynamic verification*), SWIFT achieves a substantial 30% relative improvement in speedup. Below, we analyze the factors contributing to this advantage (elaborated in Section 3.1):

- **Optimization Objective Granularity:** Self-SD calculates its optimization objective at a multi-sample level, requiring sequential decoding of all selected training samples (e.g., 8 samples with 32 tokens each) for every iteration to optimize Equation 1. In contrast,

| Methods | #Random Optimization | #Bayesian Optimization | Optimization Latency (s) | $r$ | $M$ | $\alpha$ | Speedup |
|---|---|---|---|---|---|---|---|
| SELF-SD | - | 5 | 155 | 0.50 | 1.80 | 0.57 | 0.97× |
| SELF-SD *w/ dynamic ver.* | - | 5 | 155 | 0.50 | 2.07 | 0.86 | 1.17× |
| SELF-SD$_c$ | - | 30 | 199 | 0.45 | 2.08 | 0.70 | 1.04× |
| SELF-SD$_c$ *w/ dynamic ver.* | - | 30 | 199 | 0.45 | 2.44 | 0.93 | 1.22× |
| SWIFT (Ours) | 552 | 23 | **116** | 0.45 | 5.82 | 0.98 | **1.56×** |

Table 12: Comparison of SWIFT and Self-SD at similar optimization latencies. We report the skip ratio $r$, mean generated length $M$, token acceptance rate $\alpha$, and speedup under greedy decoding. The results are obtained with LLaMA-2-13B on CNN/DM, with "*ver.*" indicating *verification*.

SWIFT adopts a step-level optimization objective, optimizing the layer set dynamically at each decoding step.

- **Bayesian Optimization Complexity:** The computational complexity of Bayesian optimization increases significantly with the number of iterations. SWIFT mitigates this burden by combining random search with interval Bayesian optimization, accelerating convergence of the optimization process while reducing computational overhead.

To further examine optimization trade-offs, we reduce Self-SD's sequential optimization requirement to a single sample with 8 tokens, enabling more Bayesian Optimization iterations within a comparable latency. The corresponding results, denoted as Self-SD$_c$ (rows 3-4), are presented in Table 12. Even with these optimized settings, SWIFT demonstrates substantial superior speedup and efficiency, highlighting the effectiveness of our proposed strategies.

## D.5 THE NECESSITY OF PLUG-AND-PLAY SD METHODS

There has been a surge of recent interest in Speculative Decoding (SD), leading to the development of numerous promising strategies in the field, which can be broadly categorized into two directions:

- **Training-required SD.** These methods require additional pretraining or fine-tuning to improve speculative accuracy, often involving the integration of extra parameters. For instance, Medusa (Cai et al., 2024) and Eagle (Li et al., 2024a;b) incorporate lightweight draft heads into target LLMs and fine-tune them, achieving 3×∼4× speedups.

- **Plug-and-play SD.** These approaches offer immediate acceleration of LLM inference without relying on auxiliary models or additional training. Notable examples include Parallel Decoding (Santilli et al., 2023) and Lookahead (Fu et al., 2024), which leverage Jacobi-based drafting, achieving 1.2×∼1.4× speedups across various LLMs.

While training-required SD methods generally deliver higher speedups, their reliance on additional training and parameters limits both their generalizability and practicality. This has sparked debate within the academic community regarding the value of plug-and-play SD methods. To address these concerns, we present a detailed analysis below to highlight the necessity of plug-and-play SD approaches and underscore the contributions of our proposed SWIFT:

**1) Training costs of training-required SD methods are often prohibitive.** Training-required methods such as Medusa (Cai et al., 2024) and Eagle (Li et al., 2024a;b), while achieving higher speedups, incur substantial training costs. Despite efforts to reduce training overhead, these methods still require extensive computational resources (e.g., GPU time and datasets) to deliver valid acceleration performance. For example, Eagle requires 1–2 days of training with 8 RTX 3090 GPUs for LLaMA-33B or up to 2 days on 4 A100 (40G) GPUs for LLaMA-2-Chat-70B, using a dataset of 70k dialogues from ShareGPT. Such computational burdens introduce challenges in several scenarios:

- Users must train new draft models for unsupported target LLMs. For example, if the user's target LLM is not among the released checkpoints or if the base model is updated (e.g.,

LLaMA-3.x), users are forced to train a new draft model, which may exceed their available GPU resources (e.g., GPU time).

- Users with small-scale acceleration needs face inefficiencies. For instance, a researcher needing to evaluate a small set of samples (e.g., 10 hours of evaluation) would find the 1–2 day training requirement disproportionate and hinder overall research efficiency.

**2) Plug-and-play SD fills critical gaps unaddressed by training-required methods.** Plug-and-play SD methods, including SWIFT, are model-agnostic and training-free, providing *immediate acceleration without requiring additional computational overhead*. These attributes are particularly critical for large models (70B–340B) and for use cases requiring rapid integration. The growing adoption of plug-and-play SD methods, such as Lookahead (Fu et al., 2024), further underscores their importance. These methods cater to scenarios where ease of use and computational efficiency are paramount, validating their research significance.

**3) SWIFT pioneers plug-and-play SD with layer-skipping drafting.** SWIFT represents the first plug-and-play SD method to incorporate *layer-skipping drafting*. It consistently achieves $1.3\times\sim1.6\times$ speedups over vanilla autoregressive decoding across diverse models and tasks. Additionally, it demonstrates $10\%\sim20\%$ higher efficiency compared to Lookahead (Fu et al., 2024). Despite its effectiveness, SWIFT introduces a **complementary research direction** for existing plug-and-play SD. Its approach is *orthogonal* to Lookahead Decoding, and combining the two could further amplify their collective efficiency. We believe this study provides valuable insights and paves the way for future SD advancements, particularly for practical and cost-effective LLM acceleration.

To sum up, while training-required SD methods achieve higher speedups, their high computational costs and limited flexibility reduce practicality. Plug-and-play SD methods, like SWIFT, offer training-free, model-agnostic acceleration, making them ideal for diverse scenarios. We hope this clarification fosters greater awareness and recognition of the value of plug-and-play SD research.

### D.6 ADDITIONAL DISCUSSIONS WITH RELATED WORK

In this work, we leverage the inherent layer sparsity of LLMs through layer skipping, which selectively bypasses intermediate layers within the target LLM to construct the compact draft model. In addition to layer skipping, there has been another research direction in SD that focuses on early exiting, where inference halts at earlier layers to improve computational efficiency (Yang et al., 2023; Hooper et al., 2023; Bae et al., 2023; Elhoushi et al., 2024). Particularly, LayerSkip (Elhoushi et al., 2024) explores early-exit drafting by generating drafts using only the earlier layers of the target LLM, followed by verification with the full-parameter model. This approach requires training involving layer dropout and early exit losses. Similarly, PPD (Yang et al., 2023) employs early exiting but trains individual classifiers for each layer instead of relying on a single final-layer classifier. Although effective, these methods rely on extensive fine-tuning to enable early-exiting capabilities, incurring substantial computational costs. Moreover, the training process alters the target LLM's original output distribution, potentially compromising the reliability of generated outputs. In contrast, our proposed SWIFT does not require auxiliary models or additional training, preserving the original output distribution of the target LLM while delivering comparable acceleration benefits.

There has been a parallel line of training-required SD research focusing on *non-autoregressive* drafting strategies (Stern et al., 2018; Cai et al., 2024; Gloeckle et al., 2024; Kim et al., 2024). These methods integrate multiple draft heads into the target LLM, enabling the parallel generation of draft tokens at each decoding step. Notably, Kim et al. (2024) builds on the Blockwise Parallel Decoding paradigm introduced in Stern et al. (2018), accelerating inference by refining block drafts with task-independent n-grams and lightweight rescorers using smaller LMs. While these approaches achieve notable acceleration, they also necessitate extensive training of draft models. SWIFT complements these efforts by pioneering plug-and-play SD that eliminates the need for auxiliary models or additional training, offering a more flexible and practical solution for diverse use cases.

### D.7 OPTIMIZATION STEPS

We present the detailed configuration of SWIFT across various optimization steps in Figure 10. As the process continues, the skipped layer set is gradually refined toward the optimal configuration.

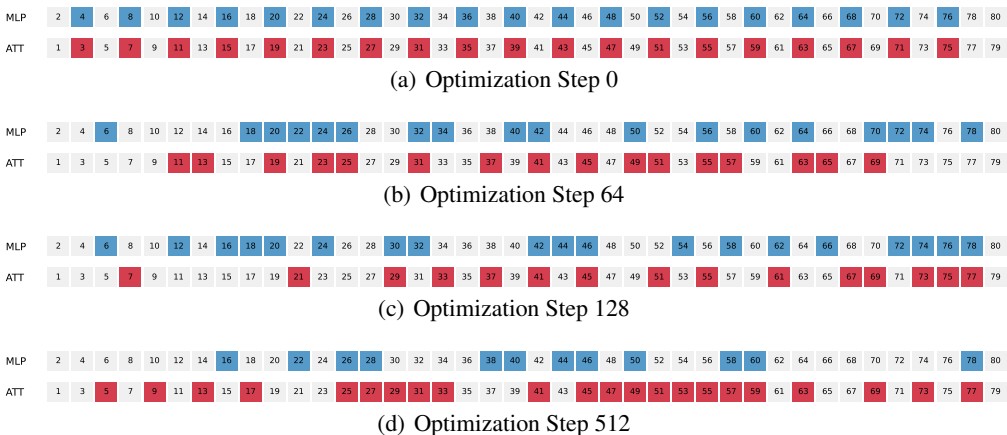

(a) Optimization Step 0

(b) Optimization Step 64

(c) Optimization Step 128

(d) Optimization Step 512

Figure 12: Visualization of skipped layer set configurations of LLaMA-2-13B optimized by SWIFT on different optimization steps. Gray squares indicate retained layers, red squares denote skipped attention layers, and blue squares signify skipped MLP layers.

## D.8 CASE STUDY

We present examples from CNN/DM and Humaneval in Table 13 and 14, respectively, comparing token acceptance between SWIFT and prior plug-and-play approach, Lookahead (Fu et al., 2024). The results demonstrate that SWIFT accepts a higher number of drafts, which are uniformly distributed in the outputs, highlighting the effectiveness of our approach.

Table 13: CNN/DM example using LLaMA-2-13B; red tokens indicate accepted drafts.

| User Input | **Article:** An Alberta student who'd accidentally left his headlights on all day was greeted by what may have been the world's friendliest note from a stranger when he returned to his car. But Derek Murray, a University ... **Summary:** |
|---|---|
| LOOKAHEAD | A Canadian law student was greeted by a friendly note from a stranger when he returned to his car after leaving the headlights on all day . |
| SWIFT | A Canadian law student was greeted by a friendly note from a stranger when he returned to his car after leaving the headlights on all day . |

Table 14: Humaneval example using CodeLLaMA-13B; red tokens indicate accepted drafts.

| User Input | from typing import List def has_close_elements(numbers: List[float], threshold: float) ->bool: """ Check if in given list of numbers, are any two numbers closer to each other than given threshold. >>>has_close_elements ... |
|---|---|
| LOOKAHEAD | for i in range(len(numbers) - 1): if abs(numbers[i] - numbers[i + 1]) <= threshold: return True return False def has_close_elements_2(numbers: List[float], threshold: float) ->bool: """ Check if in given list of numbers, are any two numbers closer to each other than given threshold. >>>has_close_elements_2 |
| SWIFT | for i in range(len(numbers) - 1): if abs(numbers[i] - numbers[i + 1]) <= threshold: return True return False def has_close_elements_2(numbers: List[float], threshold: float) ->bool: """ Check if in given list of numbers, are any two numbers closer to each other than given threshold. >>> has_close_elements_2 |

