# OpenReview forum: "SWIFT: On-the-Fly Self-Speculative Decoding for LLM Inference Acceleration"
_ICLR.cc/2025/Conference — ICLR 2025 Poster_

### Official Review · Reviewer_vnfL · 2024-10-21

**Soundness:** 2
**Presentation:** 3
**Contribution:** 2
**Rating:** 6
**Confidence:** 4

**Summary:**

This paper aims to accelerate the inference of LLMs. They introduce SWIFT, a self-speculative decoding algorithm that adaptively selects intermediate layer to skip without extra cost. They performed an empirical analysis of layer-skipping SD paradigm and show the potential of self-accelerate of LLMs through layer sparsity. They used some techniques like early-stop drafting to further speed up reasoning.

**Strengths:**

1.The paper is well-written and flows very smoothly.
2.The authors make effort to demonstrate the feasibility of their theory through experiments.
3.The method incorporates many of the latest techniques.

**Weaknesses:**

1.The author should compare their method with self-SD [1] in table 2, since their method is an improvement of the latter.
2. The author only compared to the baseline on the Llama and CodeLlama models. I believe experiments should be conducted on larger models with different architectures to demonstrate the generalization of the method.
3.Moreover, compared with self-SD, the innovation is still insufficient, for example , the confidence-aware inference strategies are similar to some mechanism in [1],[2]
4.Despite SWIFIT does not require additional training, comparing with other method ,like EAGLE [2] ,Medusa [3], which can achieve over a 3.05-4.26x speedup, SWIFIT does’t show much value.As reported in [2], the draft models is trainable within 1-2 days for 70B models.

[1] Draft & Verify: Lossless Large Language Model Acceleration via Self-Speculative Decoding. ACL 2024
[2] EAGLE-2: Faster Inference of Language Models with Dynamic Draft Trees, EMNLP 2024
[3] Medusa: Simple Framework for Accelerating LLM Generation with Multiple Decoding Heads, ICML 2024

**Questions:**

1.The author should compare their method with self-SD [1] in table 2, since their method is an improvement of the latter.
2. The author only compared to the baseline on the Llama and CodeLlama models. I believe experiments should be conducted on larger models with different architectures to demonstrate the generalization of the method.
3.Moreover, compared with self-SD, the innovation is still insufficient, for example , the confidence-aware inference strategies are similar to some mechanism in [1],[2]
4.Despite SWIFIT does not require additional training, comparing with other method ,like EAGLE [2] ,Medusa [3], which can achieve over a 3.05-4.26x speedup, SWIFIT does’t show much value.As reported in [2], the draft models is trainable within 1-2 days for 70B models.

[1] Draft & Verify: Lossless Large Language Model Acceleration via Self-Speculative Decoding. ACL 2024
[2] EAGLE-2: Faster Inference of Language Models with Dynamic Draft Trees, EMNLP 2024
[3] Medusa: Simple Framework for Accelerating LLM Generation with Multiple Decoding Heads, ICML 2024

---

> ### Author Response · Authors · 2024-11-17
> **Response - 1**
>
> We sincerely thank Reviewer vnfL for the positive feedback, and we deeply appreciate the time and effort you have dedicated to reviewing our submission. We are encouraged by the recognition of *our efforts to demonstrate the efficacy of SWIFT through experiments*. We are also delighted to know that you find our manuscript to be *well-written* and *fluid in its presentation*, and that you appreciate SWIFT's integration of *cutting-edge* techniques to enhance its practical performance.
>
> Below, we provide detailed responses to your comments:
>
>
>
> ***Q1: The author should compare their method with Self-SD[1] in table 2, since their method is an improvement of the latter.***
>
> A1: We appreciate your inquiry of comparisons between SWIFT and Self-SD [1]. We provide the comparison results below. In addition to reporting the overall speedup, we provide key metrics including the skip ratio ($r$), mean accepted tokens (*M*), and token acceptance rate ($\alpha$) for comparison. The relationship among these metrics and the expected wall-clock speedup is explained in Equation (6) of Appendix B.3.
>
> > R4-Table1: Experimental Results on CNN/DM (Greedy Decoding, FP16 Precision)
>
> | Methods         | Plug-and-Play | Optimization Latency | $r$  | *M*  | $\alpha$ |  Speedup  |
> | --------------- | :-----------: | :------------------: | :--: | :--: | :------: | :-------: |
> | Self-SD         |      No       |      ~7.2 hours      | 0.43 | 4.02 |   0.85   |   1.29x   |
> | Self-SD *w/ CA* |      No       |      ~7.2 hours      | 0.43 | 5.69 |   0.98   |   1.52x   |
> | SWIFT           |      Yes      |    **~2 minutes**    | 0.45 | 5.82 |   0.98   | **1.56x** |
>
> > *CA* refers to our proposed Confidence-aware inference Acceleration strategy in Section 4.2.
>
> Self-SD necessitates a time-intensive Bayesian Optimization process before inference (~7.5 hours for LLaMA-2-13B and ~20 hours for LLaMA-2-70B). In contrast, SWIFT introduces an on-the-fly optimization strategy, resulting in an approximate **200X reduction in optimization latency** while maintaining a **1.56x speedup**. We further augmented Self-SD with our *Confidence-aware inference Acceleration strategy* (Self-SD *w/ CA*). Even compared to this augmented version, SWIFT achieves competitive speedups.
>
> We provide further comparative analysis of SWIFT versus Self-SD in **A2&A3 to Reviewer tWD9 (R2)**, discussing speedups, computational overhead, and performance with limited optimization iterations. These results and discussions will be incorporated into the revised manuscript. We sincerely appreciate your inquiry, which allowed us to strengthen the comparative analysis of our work.
>
>
>
> [1] Draft & Verify: Lossless Large Language Model Acceleration via Self-Speculative Decoding. Zhang et.al. ACL 2024.

---

> ### Author Response · Authors · 2024-11-17
> **Response - 2**
>
> ***Q2: The author only compared to the baseline on the Llama and CodeLlama models. I believe experiments should be conducted on larger models with different architectures to demonstrate the generalization of the method.***
>
> A2: We would like to clarify that **we have already included the experimental results for Yi-34B and DeepSeek-Coder-33B**, along with their instruction-tuned variants in our paper, as presented in Figure 9. The detailed experimental results are illustrated in Appendix C.2. The results indicate that SWIFT achieves **efficiency improvements** ranging from **26% to 54%** on these LLM backbones, which substantiates the generalized utility of SWIFT as a general-purpose, plug-and-play SD method, offering promising inference acceleration across diverse LLM backbones.
>
> To further demonstrate the generalization ability of SWIFT, **we have consolidated all results related to diverse LLM backbones below** for your convenience. If you have further suggestions for additional backbones to evaluate, please feel free to propose them, and we will gladly incorporate these in our future analyses.
>
> > R4-Table2: Experimental Results on CNN/DM (Greedy Decoding, FP16 Precision)
>
> | Models                  | $r$  | *M*  | $\alpha$ | Speedup |
> | ----------------------- | :--: | :--: | :------: | :-----: |
> | LLaMA-2-70B             | 0.50 | 3.85 |   0.99   |  1.43x  |
> | LLaMA-2-70B-Chat        | 0.50 | 3.43 |   0.85   |  1.31x  |
> | LLaMA-3-70B             | 0.40 | 5.43 |   0.99   |  1.41x  |
> | LLaMA-3-70B-Instruct    | 0.40 | 3.76 |   0.95   |  1.33x  |
> | CodeLLaMA-34B           | 0.50 | 3.79 |   0.88   |  1.46x  |
> | Yi-34B                  | 0.45 | 2.74 |   0.94   |  1.30x  |
> | Yi-34B-Chat             | 0.45 | 2.84 |   0.91   |  1.29x  |
> | DeepSeek-Coder          | 0.50 | 4.97 |   0.99   |  1.54x  |
> | DeepSeek-Coder-Instruct | 0.50 | 3.80 |   0.88   |  1.39x  |
>
> These results further validate **SWIFT’s generalization ability** across a broader range of LLMs, including both chat and instruction-tuned variants.

---

> ### Author Response · Authors · 2024-11-17
> **Response - 3 (1)**
>
> ***Q3: Compared with Self-SD, the innovation is still insufficient, for example, the confidence-aware inference strategies are similar to some mechanism in Self-SD[1] and EAGLE[2].***
>
> A3: Thank you for raising this question regarding the innovations of SWIFT compared to Self-SD [1]. Below, we provide a detailed explanation to address this concern:
>
> ------
>
> **(1) Self-SD necessitates substantial optimization latency, making it unsuitable for plug-and-play LLM inference acceleration.**
>
> As detailed in Section 3 (L148–174) of our manuscript, Self-SD was the first work to explore layer-skipping drafting within the Speculative Decoding (SD) paradigm, which proposes utilizing a Bayesian Optimization process before inference to determine the skipped layer set for efficienct drafting. While this method shows promising efficacy, it nessisitates **substantial computational overhead and optimization latency** (~7.5 hours for LLaMA-2-13B and ~20 hours for LLaMA-2-70B), rendering it **unsuitable for plug-and-play LLM inference acceleration scenarios**.
>
> To further illustrate Self-SD's **optimization latency**, we conducted an experiment varying the number of Bayesian optimization iterations (Self-SD uses 1000 iterations by default). The results are shown below:
>
> > R4-Table3: Experimental Results on LLaMA-2-13B, CNN/DM (Greedy Decoding, FP16 Precision)
>
> | #Bayesian_Opt  | Optimization Latency (s) | $r$  | *M*  | $\alpha$ | Speedup |
> | :------------: | :----------------------: | :--: | :--: | :------: | :-----: |
> |       0        |            0             | 0.50 | 1.75 |   0.56   |  0.96x  |
> |       10       |           279            | 0.49 | 1.83 |   0.57   |  0.97x  |
> |       50       |           1474           | 0.49 | 1.80 |   0.61   |  1.02x  |
> |      100       |           2898           | 0.45 | 3.04 |   0.80   |  1.19x  |
> |      200       |           5517           | 0.48 | 3.47 |   0.84   |  1.24x  |
> | 1000 (default) |          27071           | 0.43 | 4.02 |   0.85   |  1.29x  |
>
> From the table, we observe that Self-SD achieves **negligible speedup improvement** with **fewer than 50** Bayesian optimization iterations (nearly equivalent to unified skipping, *#Bayesian Opt = 0*). At 100 iterations, Self-SD achieves a 1.19x speedup, but its optimization latency is nearly **25 times** that of SWIFT (1 hour *vs.* 2 minutes).
>
> To further evaluate Self-SD’s performance under **the plug-and-play requirement** (i.e., optimization latency under 2 minutes), we conducted additional experiments:
>
> > R4-Table4: Experimental Results on LLaMA-2-13B, CNN/DM (Greedy Decoding, FP16 Precision)
>
> | Methods                           | #Bayesian_Opt |    Opt_Time (s)    | $r$  | *M*  | $\alpha$ | Speedup |
> | --------------------------------- | :-----------: | :----------------: | :--: | :--: | :------: | :-----: |
> | Self-SD (default)                 |     1000      | 27071 (~7.5 hours) | 0.43 | 4.02 |   0.85   |  1.29x  |
> | Self-SD (for plug-and-play usage) |       5       |  155 (~ 2.5 mins)  | 0.50 | 1.80 |   0.57   |  0.97x  |
>
> Under the plug-and-play constraint (optimization latency < 2 minutes), Self-SD's speedup effect **drops significantly**, resulting in a **negative acceleration speedup (0.97x)**. This demonstrates that the substantial optimization overhead of Self-SD makes it ***an invalid solution*** for plug-and-play LLM inference acceleration and highlights ***the great challenges*** to develop plug-and-play SD methods with layer-skipping drafting.
>
> ------
>
> **(2) We propose the first plug-and-play layer-skipping SD method, introducing significant innovations to the layer set optimization strategy.**
>
> Below, we detail **the contributions of SWIFT over Self-SD**, as discussed in L168–L174 & Section 4 of our manuscript:
>
> - **Optimization Objective Granularity:** Self-SD calculates its optimization objective at a multi-sample level, requiring sequential decoding of all selected training samples (e.g., 8 samples with 32 tokens each) for every iteration to optimize Equation (1). In contrast, SWIFT adopts a **step-level optimization objective**, dynamically optimizing the layer set at each decoding step, which significantly reduces computational overhead.
> - **Bayesian Optimization Complexity:** The computational complexity of Bayesian optimization grows substantially with the number of iterations. SWIFT mitigates this burden by combining **random search** with **interval Bayesian optimization**, which accelerates convergence while reducing the overall computational complexity of the optimization process.
>
> These innovations enable SWIFT to optimize the skipped layer set of the target LLM **on the fly**, delivering LLM inference acceleration as a **plug-and-play SD solution**. Additionally, as you noted, SWIFT seamlessly integrates multiple advanced SD techniques, including tree drafting and confidence-aware candidate pruning, further enhancing its efficiency for practical inference acceleration.
>
> ------

---

> ### Author Response · Authors · 2024-11-17
> **Response - 3 (2)**
>
> **(3) Our proposed SWIFT obtains a promising 1.3x-1.6x speedup as a plug-and-play SD method, competitive with Self-SD which requires substantial optimization demands.**
>
> > R4-Table5: Experimental Results on LLaMA-2-13B, CNN/DM (Greedy Decoding, FP16 Precision)
>
> | Methods         | #Bayesian_Opt | Optimization Latency | $r$  | *M*  | $\alpha$ |  Speedup  |
> | --------------- | :-----------: | :------------------: | :--: | :--: | :------: | :-------: |
> | Self-SD         |     1000      |      ~7.2 hours      | 0.43 | 4.02 |   0.85   |   1.29x   |
> | Self-SD *w/ CA* |     1000      |      ~7.2 hours      | 0.43 | 5.69 |   0.98   |   1.52x   |
> | Self-SD         |       5       |     ~2.5 minutes     | 0.50 | 1.80 |   0.57   |   0.97x   |
> | Self-SD w/ CA   |       5       |     ~2.5 minutes     | 0.50 | 2.07 |   0.86   |   1.17x   |
> | SWIFT           |               |    **~2 minutes**    | 0.45 | 5.82 |   0.98   | **1.56x** |
>
> > *CA* refers to our proposed Confidence-aware inference Acceleration strategy in Section 4.2.
>
> As shown in table 5, Self-SD requires a computationally expensive Bayesian Optimization process before inference (~7.5 hours for LLaMA-2-13B and ~20 hours for LLaMA-2-70B), which makes it unsuitable for plug-and-play applications. In contrast, SWIFT’s **on-the-fly optimization strategy** achieves an approximate **200x reduction in optimization latency**, with an impressive **1.56x speedup** in inference performance.
>
> To validate our approach further, we augmented Self-SD with the *Confidence-aware Inference Acceleration Strategy* (*Self-SD w/ CA*). Even with this enhancement, SWIFT demonstrates competitive or superior performance, achieving higher speedups while maintaining minimal latency overhead.
>
> ------
>
> **(4) As the first plug-and-play SD method with layer-skipping drafting, we hope SWIFT provides valuable insights and inspires further research in this area.**
>
> Speculative Decoding (SD) has recently garnered significant interest from both academia and industry as an effective LLM inference acceleration strategy that preserves the original LLM's output distribution. It has been widely adopted in LLM inference applications, such as [vLLM](https://github.com/vllm-project/vllm). However, recent SD research appears to have reached a plateau, focusing mainly on incremental improvements or revisiting prior methods without fresh, innovative explorations in this field.
>
> In this work, we present **the first exploration of plug-and-play SD methods with layer-skipping drafting**. SWIFT introduces an **orthogonal** approach to Lookahead Decoding [2], showcasing **promising adaptability** across diverse LLMs and dynamic data streams. Unlike most existing SD methods, SWIFT operates ***without the need for auxiliary models or additional training***, making it both **cost-effective** and **practical** for real-world applications. We believe this study not only paves the way for new research directions within the community but also provides substantial value for low-cost deployment.
>
> ------
>
> **To sum up:**
>
> In contrast to Self-SD, which incurs substantial optimization latency, we propose SWIFT — **the first plug-and-play layer-skipping  SD method capable of dynamically optimizing the skipped layer set on the fly**. The efficiency supriority of SWIFT is established upon two key innovations regarding *optimization objective granularity* and *Bayesian optimization efficiency*. These advancements allow SWIFT to perform layer set optimization within user-defined acceleration tolerances, resulting in a remarkable **200x reduction in optimization latency** compared to Self-SD. We believe this study opens new avenues for research in the community while offering substantial value for practical and low-cost deployment.
>
>
>
> [1] Draft & Verify: Lossless Large Language Model Acceleration via Self-Speculative Decoding. Zhang et.al. ACL 2024.
>
> [2] Break the Sequential Dependency of LLM Inference Using Lookahead Decoding. Fu et.al. ICML 2024.

---

> ### Author Response · Authors · 2024-11-17
> **Response - 4**
>
> ***Q4: Despite SWIFT does not require additional training, comparing with other method, like EAGLE[2], Medusa[1], which can achieve over a 3.05-4.26x speedup, SWIFIT does’t show much value. As reported in EAGLE[2], the draft models is trainable within 1-2 days for 70B models.***
>
> A4: Thank you for raising this important question regarding the value of plug-and-play methods like SWIFT compared to training-intensive approaches such as Medusa [1] and EAGLE [2]. Below, we address your concern in detail:
>
> ------
>
> **(1) The computational overhead of training-required methods is unacceptable sometimes.**
>
> Training-required methods such as Medusa [1] and EAGLE [2], while achieving higher speedups, still incur **substantial training costs**. Despite efforts to reduce training overhead, these methods require extensive computational resources (e.g., GPU time and datasets) to deliver valid acceleration performance. For example: EAGLE requires **1–2 days of training with 8 RTX 3090 GPUs** for LLaMA-33B or **up to 2 days on 4 A100 (40G) GPUs** for LLaMA2-Chat-70B, utilizing a dataset of **70k dialogues** from ShareGPT.
>
> These computational burdens introduce challenges in several scenarios:
>
> - **Users must train new draft models for unsupported target LLMs.** If the user's target LLM is not among EAGLE's released checkpoints or if the base model is updated (e.g., LLaMA-3.x), users are forced to train a new draft model, which may exceed their available GPU resources (e.g., GPU time).
> - **Users with small-scale acceleration needs face inefficiencies.** For instance, a researcher needing to evaluate a small set of samples (e.g., 10 hours of evaluation) would find the 1–2 day training requirement for EAGLE disproportionate and harmful to overall research efficiency.
>
> ------
>
> **(2) High speedups in training-required methods do not negate the value of plug-and-play SD research.**
>
> Plug-and-play SD methods, including SWIFT, are **model-agnostic and training-free**, providing **immediate acceleration without requiring additional computational overhead**. These attributes are particularly critical for large models (70B–340B) and specific use cases, as discussed above.
>
> Additionally, the increasing adoption of plug-and-play SD methods such as Lookahead [3] and PLD [4] (supported in [vLLM](https://github.com/vllm-project/vllm)) highlights the demand for ready-to-use solutions. This further **validates the research value of plug-and-play SD methods**, which cater to scenarios where computational efficiency and ease of integration are paramount.
>
> ------
>
> **(3) SWIFT pioneers plug-and-play SD with layer-skipping drafting, achieving state-of-the-art performance.**
>
> As detailed in A2, SWIFT represents the **first plug-and-play SD method** to incorporate **layer-skipping drafting**. It consistently achieves **1.3x–1.6x speedups** over vanilla autoregressive decoding across diverse models and tasks. Additionally, it demonstrates **10%–20% higher efficiency** compared to Lookahead Decoding [3].
>
> Beyond its promising experimental results, SWIFT introduces a **novel and complementary research direction** for plug-and-play SD methods. Its approach is **orthogonal** to Lookahead Decoding, and combining the two could further amplify their collective efficiency. We believe this study provides valuable insights and paves the way for future advancements in the SD community, particularly for practical and cost-effective LLM acceleration.
>
>
>
> [1] Medusa: Simple LLM Inference Acceleration Framework with Multiple Decoding Heads. Cai et.al. ICML 2024.
>
> [2] EAGLE: Speculative Sampling Requires Rethinking Feature Uncertainty. Li et.al. ICML 2024.
>
> [3] Break the Sequential Dependency of LLM Inference Using Lookahead Decoding. Fu et.al. ICML 2024.
>
> [4] Prompt Lookup Decoding. Apoorv Saxena. 2023. [Github Repository](https://github.com/apoorvumang/prompt-lookup-decoding/).

---

> > ### Comment · Reviewer_vnfL · 2024-11-26
> > **Response**
> >
> > Thank you for your response, which has helped me better understand the strengths of this paper. I will increase my score.

---

> ### Author Response · Authors · 2024-11-17
> **Response - 5**
>
> **To summarize and for further discussion:**
>
> - In the responses above, we have provided the **additional experimental results** you suggested (Q1 & Q2), which further illustrate the effectiveness of SWIFT. If you have any **additional comments** regarding the robustness of our approach—such as suggestions for new experiments or alternative interpretations of results—**please feel free to share them**. We would greatly appreciate the opportunity to engage further and address any remaining concerns.
>
> - We also acknowledge your concerns regarding **the value of plug-and-play SD research** and **the innovations introduced by SWIFT** (Q3 & Q4). **We have provided detailed responses to address these points.** Recognizing the value of a research direction is indeed ***a serious matter***, and we deeply respect your perspective. If you still have reservations about the value of pursuing plug-and-play SD research, we encourage you to share them with us. Your feedback would be invaluable in helping us rethink and refine our future research trajectory.
> - After addressing your suggested experiments and elaborating on the significance of plug-and-play SD methods, **would you reconsider your current rating?** If you decide not to adjust your rating, we would be grateful if you could clarify whether this decision stems from concerns about unresolved experimental issues or a lack of confidence in the research direction itself.
>
> We look forward to continuing the discussion with you. Thank you once again for the time and effort you have dedicated to reviewing our submission. Your insights are greatly appreciated.

---

> ### Author Response · Authors · 2024-11-23
> **Follow-Up: Seeking Further Feedback**
>
> Dear Reviewer, I hope you're doing well. Following up on our recent exchange regarding this paper, I wanted to check if there are any further concerns or feedback from your side. Your insights are invaluable to us, and we're keen to address any remaining issues.

---

> ### Author Response · Authors · 2024-11-25
> **Kindly Reminder on the Discussion Period**
>
> Dear Reviewer, I hope this message finds you well. As the discussion period is nearing its end with **only two days remaining**, I wanted to ensure we have addressed all your concerns satisfactorily. If there are any additional points or feedback you'd like us to consider, please let us know. Your insights are invaluable to us, and we’re eager to address any remaining issues to improve our work.
>
> Thank you for your time and effort in reviewing our paper.

---

### Official Review · Reviewer_LGVh · 2024-11-03

**Soundness:** 3
**Presentation:** 3
**Contribution:** 2
**Rating:** 5
**Confidence:** 4

**Summary:**

This paper proposes a plug-and-play self-speculative decoding method. The authors employ a layer-skipping approach to construct a draft model. Experimental results indicate that this method achieves a 1.3-1.6 times inference speedup on Llama-2 and Code-Llama models.

**Strengths:**

The method does not require training an additional model or module for drafting, making it applicable to most large language models.

**Weaknesses:**

1. The speedup is not as promising compared to other training-free methods like Lookahead. The authors should also present results for methods such as Mudusa and Eagle, which require minimal training overhead.
2. It is recommended that the authors test well-trained LLMs, such as Llama-3, as models with less effective performance might yield higher speedup ratios.
3. The method requires different settings for different tasks. However, in real-world LLM chat applications, it is often difficult to predict the corresponding tasks of user instructions. It is suggested that the authors evaluate the method's speedup performance on benchmarks like MT-Bench, which test the general capabilities of models.

**Questions:**

1. Table 2 presents results for Llama-2-13B, Llama-2-13B-Chat, and Llama-2-70B. Why are the results for Llama-2-70B-Chat and Llama-2-7B(-Chat) not included?
2. How does the overhead of the proposed layer searching algorithm compare to the overhead of training additional modules like Eagle?

---

> ### Author Response · Authors · 2024-11-17
> **Response - 1**
>
> We are grateful for the time Reviewer LGVh have spent reviewing our submission. We appreciate your recognition of the strengths of our proposed SWIFT, particularly its ability to accelerate LLM inference *without introducing additional model parameters or modules for drafting*, thereby ensuring *broad applicability* across various LLMs.
>
> That said, we would like to **address some important misunderstandings** that may have affected the evaluation of our work. Specifically, there appears to be confusion regarding the performance comparison with Lookahead (as detailed in Table 2 of our main results) and the training overhead associated with training-required methods versus plug-and-play approaches. These points are crucial for accurately assessing the significance of our contributions.
>
> Below, we provide detailed clarifications for each of your comments.
>
>
>
> ***Q1: The speedup is not as promising compared to other training-free methods like Lookahead.***
>
> A1: **This appears to be a significant misunderstanding, as we have already provided a detailed comparison with Lookahead [1] in Table 2 (main results) of our manuscript.** As shown in Table 2, SWIFT consistently achieves superior efficiency compared to prior training-free methods, including Lookahead and Parallel Decoding [2]. Specifically:
>
> - SWIFT achieves speedups of **1.3x–1.6x** over vanilla autoregressive decoding across various models and tasks.
> - It delivers **10%–20%** higher efficiency compared to Lookahead Decoding.
>
> Additionally, Appendix D.4 (Tables 9 and 10) presents **detailed token acceptance comparisons**, further underscoring SWIFT's advantages over Lookahead.
>
> Beyond performance metrics, as discussed in Section 1 (L49–L60, Figure 1) and Section 2 (L126–133), SWIFT introduces **sparsity-based drafting**, a novel and **complementary research direction** for plug-and-play speculative decoding (SD). These directions, summarized and visualized in Figure 1, are as follows:
>
> - **Jacobi-based drafting (prior methods):** This approach appends multiple pseudo tokens to the input prompt, allowing the LLM to generate several tokens as drafts in a single step.
> - **Sparsity-based drafting (ours):** SWIFT leverages the inherent layer sparsity within LLMs to enable efficient drafting by adaptively optimizing the set of skipped layers during inference.
>
> These two approaches are **orthogonal and complementary**, and combining them could amplify the efficiency of both. For instance, SWIFT could incorporate a Lookahead-like mechanism during drafting, which is expected to further enhance both drafting efficiency and token acceptance rates.
>
> To the best of our knowledge, SWIFT is the **first approach to explore plug-and-play SD using sparsity-based drafting**. We hope our findings provide valuable insights and inspire further research in this area.
>
>
>
> [1] Break the Sequential Dependency of LLM Inference Using Lookahead Decoding. Fu et.al. ICML 2024.
>
> [2] Accelerating Transformer Inference for Translation via Parallel Decoding. Santilli et.al. ACL 2023.

---

> ### Author Response · Authors · 2024-11-17
> **Response - 2**
>
> ***Q2: The authors should also present results for methods such as Mudusa and Eagle, which require minimal training overhead. How does the overhead of the proposed layer searching algorithm compare to the overhead of training additional modules like Eagle?***
>
> A2: **In fact, training-required methods such as Medusa [1] and Eagle [2] still incur substantial training costs.** We did not explicitly discuss these costs in our manuscript because the training-cost gap between training-required SD methods and plug-and-play SD methods is already well recognized in the field.
>
> To further highlight SWIFT’s efficiency, we provide a detailed breakdown of the training and optimization costs for these methods (refer to *R2-Table2* for additional details):
>
> > R3-Table1: Comparison of training and optimization costs for Llama-2-13B
>
> | Methods                  | Eagle                         | LayerSkip                                | Self-SD                                                | SWIFT                           |
> | ------------------------ | :---------------------------- | :--------------------------------------- | :----------------------------------------------------- | :------------------------------ |
> | **Training Cost**        | 1-2 days with 8 RTX 3090 GPUs | 50k training steps with 64 A100s (80 GB) | 1000 Bayesian Optimization Iterations Before inference | **N/A**                         |
> | **Optimization Latency** | -                             | -                                        | ~7.2 hours                                             | **~2 minutes (200x reduction)** |
>
> Detailed comparisons:
>
> - **Compared to training-required methods:** We compare the training and optimization costs of SWIFT with two representative training-required methods -- Eagle [2] and LayerSkip [3], which necessitate a time-intensive fine-tuning process on a large amount of data. In contrast, SWIFT is a *plug-and-play* SD solution that is applicable to most LLMs **without requiring additional training** and offers **immediate usability** for accelerating LLM inference.
> - **Compared to Self-SD:** Self-SD [4] involves an extensive Bayesian Optimization process before inference, which introduces significant latency (e.g., ~7.5 hours for LLaMA-2-13B, ~20 hours for LLaMA-2-70B). SWIFT introduces an **on-the-fly optimization** strategy, reducing optimization latency by approximately **200x** while maintaining 1.3x–1.6x speedups over vanilla autoregressive decoding.
>
> These comparisons underscore SWIFT’s superior efficiency in terms of both training and optimization costs.
>
> ------
>
> **The Necessity of Plug-and-Play SD Methods:**
>
> Additionally, we here further discuss the nessisity of plug-and-play methods for Speculative Decoding (SD):
>
> While training-required methods (e.g., Medusa [1], Eagle [2]) effectively push the boundaries of SD efficiency by incorporating lightweight draft modules and aligning them with target LLMs, they still demand substantial computational resources (e.g., GPU time, datasets) to deliver meaningful acceleration.
>
> For example, Eagle [2], the current SOTA SD method, provides fine-tuned checkpoints for only 11 models across 5 LLM series in its public [repository](https://github.com/SafeAILab/EAGLE). Users have to train new checkpoints on their own if:
>
> - Their target LLM is not among the released checkpoints.
> - The LLM base is updated (e.g., LLaMA-3.x series).
>
> In contrast, **plug-and-play SD methods, such as SWIFT, are model-agnostic and training-free**, offering immediate acceleration without computational overhead. This is particularly valuable for large-scale models (70B–340B), where retraining/fine-tuning is often infeasible. The widespread adoption of plug-and-play SD methods like Lookahead [5] and PLD [6] (supported in vLLM) highlights the demand for ready-to-use solutions, especially in settings like local LLM inference and online API services.
>
> ------
>
> We hope these comparisons and insights help clarify SWIFT’s contributions and practical value as an innovative plug-and-play SD method. We will incorporate these results and discussions into our revised manuscript. Thank you again for your feedback.
>
>
>
> [1] Medusa: Simple LLM Inference Acceleration Framework with Multiple Decoding Heads. Cai et.al. ICML 2024.
>
> [2] EAGLE: Speculative Sampling Requires Rethinking Feature Uncertainty. Li et.al. ICML 2024.
>
> [3] Layer Skip: Enabling Early Exit Inference and Self-Speculative Decoding. Elhoushi et.al. ACL 2024.
>
> [4] Draft & Verify: Lossless Large Language Model Acceleration via Self-Speculative Decoding. Zhang et.al. ACL 2024.
>
> [5] Break the Sequential Dependency of LLM Inference Using Lookahead Decoding. Fu et.al. ICML 2024.
>
> [6] Prompt Lookup Decoding. Apoorv Saxena. 2023. [Github Repository](https://github.com/apoorvumang/prompt-lookup-decoding/).

---

> ### Author Response · Authors · 2024-11-17
> **Response - 3**
>
> ***Q3: It is recommended that the authors test well-trained LLMs, such as Llama-3, as models with less effective performance might yield higher speedup ratios.***
>
> A3: In our manuscript, we evaluated the LLaMA-2 series following the experimental settings of Lookahead [1] and Self-SD [2], ensuring **fair comparisons with prior work**. To address your concern, we conducted additional experiments comparing SWIFT’s speedup performance on the LLaMA-2 and LLaMA-3 series, thereby showcasing **the robustness of SWIFT** regardless of the model’s overall effectiveness.
>
> In addition to reporting the overall speedup, we provide key metrics including the skip ratio ($r$), mean accepted tokens (*M*), and token acceptance rate ($\alpha$) for comparison. The relationship among these metrics and the expected wall-clock speedup is explained in Equation (6) of Appendix B.3.
>
> > R3-Table1: Experimental Results on CNN/DM (Greedy Decoding, FP16 Precision)
>
> | Models      | $r$  | *M*  | $\alpha$ | Speedup |
> | ----------- | :--: | :--: | :------: | :-----: |
> | LLaMA-2-7B  | 0.40 | 3.45 |   0.94   |  1.24x  |
> | LLaMA-3-8B  | 0.40 | 3.80 |   0.93   |  1.25x  |
> | LLaMA-2-70B | 0.50 | 3.85 |   0.99   |  1.43x  |
> | LLaMA-3-70B | 0.40 | 5.43 |   0.99   |  1.41x  |
>
> > During the optimization phase, the layer skip ratio ($r$) for LLaMA-3-70B was automatically adjusted from 0.5 to 0.4, as the token acceptance rate ($\alpha$) remained below the predefined tolerance threshold (e.g., 0.7). The adjusted ratio is reflected in the table above.
>
> These results demonstrate that SWIFT consistently achieves significant speedups (**1.2x–1.4x**) across both LLaMA-2 and LLaMA-3 series, effectively addressing the assumption that *"models with less effective performance might yield higher speedup ratios."* Although differences in layer redundancy are observed between models (e.g., $r$ values for LLaMA-2-70B vs. LLaMA-3-70B), SWIFT remains robust and adaptable, maintaining high acceleration performance irrespective of the model’s effectiveness.
>
> We sincerely appreciate your suggestion, as it allowed us to strengthen the comparative analysis of SWIFT. These additional results and discussions will be incorporated into our revised manuscript.
>
> [1] Break the Sequential Dependency of LLM Inference Using Lookahead Decoding. Fu et.al. ICML 2024.
>
> [2] Draft & Verify: Lossless Large Language Model Acceleration via Self-Speculative Decoding. Zhang et.al. ACL 2024.

---

> ### Author Response · Authors · 2024-11-17
> **Response - 4**
>
> ***Q4: The method requires different settings for different tasks. However, in real-world LLM chat applications, it is often difficult to predict the corresponding tasks of user instructions. It is suggested that the authors evaluate the method's speedup performance on benchmarks like MT-Bench, which test the general capabilities of models.***
>
> A4: We appreciate your inquiry regarding SWIFT's performance across diverse data types. Actually, as we demonstrate in Figure 2 (Section 3.2.1), ***SWIFT can achieve an average 1.2x speedup even without any task-specific optimization*** by using a unified layer skipping pattern. Building on this foundation, SWIFT is designed to dynamically optimize its acceleration performance by adjusting to the characteristics of the current data stream. As discussed in Section 5.2 (Line 462) and further elaborated in *A6 to R2*, ***SWIFT’s efficiency improves as input length and the number of instances increase***.
>
> This dynamic optimization mechanism makes SWIFT particularly effective in scenarios with large volumes of homogeneous data from specific tasks (e.g., specific test set) —common in both research and industrial applications. Furthermore, SWIFT accommodates application scenarios where user prompts exhibit *inertia*—that is, users often ask similar types of questions consecutively.
>
> Besides, in real-world LLM applications, user prompts can often be clustered into similar categories. For instance, MT-Bench organizes its data into 8 task types, representing diverse user needs. In such scenarios, a promising enhancement for SWIFT could involve **caching optimal settings** for each task type and dynamically retrieving the corresponding layer configuration to accelerate inference for incoming data. While this remains a promising direction for future exploration, it underscores SWIFT's potential to effectively handle the challenges posed by real-world applications.
>
> Additionally, as shown in **Figure 7**, SWIFT demonstrates **robustness to domain shifts and varying data types**, which contrasts with prior methods like Self-SD [1] that are sensitive to such variations and struggle to handle different data types. This adaptability further highlights SWIFT’s superiority over existing layer-skipping SD approaches.
>
> We sincerely appreciate your feedback and believe this explanation demonstrates SWIFT’s strengths and versatility, including its ability to adapt to dynamic input data streams and real-world LLM applications.
>
>
>
> [1] Draft & Verify: Lossless Large Language Model Acceleration via Self-Speculative Decoding. Zhang et.al. ACL 2024.

---

> ### Author Response · Authors · 2024-11-17
> **Response - 5**
>
> ***Q5: Table 2 presents results for Llama-2-13B, Llama-2-13B-Chat, and Llama-2-70B. Why are the results for Llama-2-70B-Chat and Llama-2-7B(-Chat) not included?***
>
> A5: Thank you for your inquiry regarding additional experimental results. In response, we now provide results for **LLaMA-2-70B-Chat** and **LLaMA-3-70B-Instruct**, complementing the previously presented results for LLaMA-2-13B, LLaMA-2-13B-Chat, and LLaMA-2-70B, as well as those provided in *R3-Table1*.
>
> > R3-Table2: Experimental Results on CNN/DM (Greedy Decoding, FP16 Precision)
>
> | Models               | $r$  | *M*  | $\alpha$ | Speedup |
> | -------------------- | :--: | :--: | :------: | :-----: |
> | LLaMA-2-70B-Chat     | 0.5  | 3.43 |   0.85   |  1.31x  |
> | LLaMA-3-70B-Instruct | 0.4  | 3.76 |   0.95   |  1.33x  |
>
> These additional results further validate SWIFT’s effectiveness across a broader range of LLaMA models, including both chat-tuned and instruction-tuned variants.
>
> We appreciate your suggestion and will incorporate these additional comparisons and discussions into our revised manuscript.
>
>
>
> **To summarize:**
>
> In the discussion above, we have addressed key misunderstandings (**A1, A2**), provided additional experimental results for the LLaMA-3 series and LLaMA-2-70B-Chat to further substantiate our claims (**A3, A5**), and offered a detailed response to your inquiry regarding SWIFT's performance across diverse data types (**A4**).
>
> Upon reviewing the overall comments, we observe that **there are no direct challenges to the core idea of our work**. Specifically, we have clarified the overlooked comparisons with Lookahead and highlighted the substantial training costs associated with training-required methods like Eagle. **These clarifications further strengthen the motivation behind SWIFT and validate its effectiveness.**
>
> **Given that the weaknesses raised by the reviewer are largely based on misunderstandings, we respectfully hope that you will engage in a thorough discussion of our clarifications.** Additionally, we kindly request that you reconsider your rating in light of the responses and evidence we have provided. If you have any further questions or require additional clarification, please feel free to let us know.

---

> ### Author Response · Authors · 2024-11-23
> **Follow-Up: Seeking Further Feedback**
>
> Dear Reviewer, I hope you're doing well. Following up on our recent exchange regarding this paper, I wanted to check if there are any further concerns or feedback from your side. Your insights are invaluable to us, and we're keen to address any remaining issues.

---

> > ### Comment · Reviewer_LGVh · 2024-11-24
> >
> > Some of my concerns have been addressed, but I will keep my score for the following reasons:
> >
> > 1. The authors claim that the proposed SWIFT can only achieve a 1.2x speedup without task-specific optimization, which is a common setting for real-world LLM chat applications. On average, Lookahead also achieves a 1.2x speedup without any task-specific optimization. This indicates that, under the common setting, SWIFT can not outperform Lookahead.
> >
> > 2. I am still wondering about SWIFT's performance on benchmarks, such as MT-Bench, that evaluate general capabilities.

---

> ### Author Response · Authors · 2024-11-24
> **Response to Reviewer LGVh**
>
> Thanks for your prompt response. We are glad to hear that some of your concerns have been addressed. And we appreciate your articulation of the reasons for maintaining your current score. However, we believe there are still some **critical misunderstandings** in your evaluation. Below, we provide further clarifications and additional experiments to address them comprehensively.
>
> ------
>
> **1.Clarifications on Key Misunderstandings**
>
> **(a) SWIFT without optimization is only a naive baseline.**
>
>
> The "*SWIFT without optimization – using a unified layer-skipping pattern*" represents only a **naive baseline** and serves as the starting point in Figure 6. In contrast to this static setting, **the core innovation** of SWIFT lies in its ability to **dynamically optimize the skipped layer configuration on the fly**. As shown in Figure 6, this optimization process rapidly improves both matchness scores and instance speedup **within the first few decoding steps**, significantly outperforming the static baseline. The optimization continues to enhance speedup throughout the inference process.
>
> **(b) Misinterpretation of SWIFT’s applicability to real-world applications.**
>
> There seems to be another misunderstanding that "SWIFT *could not perform any optimization* in real-world LLM chat applications." To address this, we emphasize that **a key innovation in SWIFT** is its ability to perform skipped layer optimization **at the step level**. Even during inference with a single input instance, SWIFT can perform optimization at early LLM decoding steps, adapting to the current instance and improving upon the static unified skipping pattern. This step-level optimization mechanism ensures SWIFT's applicability across general inference cases, including real-world chat-model applications and domain-specific tasks.
>
> **(c) Complementarity with Lookahead Decoding.**
>
> It is important to note that SWIFT is an **orthogonal and complementary method** to Lookahead Decoding [1]. The two approaches can be combined to amplify their respective efficiencies. Furthermore, as you recognized, even our starting point (the naive baseline) achieves comparable efficiency to Lookahead. With our proposed optimization mechanism, SWIFT delivers **10%–20% higher efficiency gains** compared to Lookahead.
>
> ------
>
> **2.Additional comparisons on MT-Bench**
>
> To address your concerns regarding SWIFT’s performance on other benchmarks, we conducted additional evaluations on MT-Bench using Vicuna-v1.3, a widely adopted LLM for chat applications. The results are as follows:
>
> > R3-2-Table1: Experimental results on Vicuna-7B-v1.3 (Greedy Decoding, FP16 Precision)
>
> | Methods   |  Writing  | Roleplay  | Reasoning |   Math    |  Coding   | Extraction |   Stem    | Humanities |  Overall  |
> | --------- | :-------: | :-------: | :-------: | :-------: | :-------: | :--------: | :-------: | :--------: | :-------: |
> | Lookahead |   1.07x   |   1.12x   |   1.09x   |   1.21x   |   1.17x   |   1.14x    |   1.12x   |   1.15x    |   1.13x   |
> | SWIFT     | **1.22x** | **1.27x** | **1.23x** | **1.22x** | **1.28x** | **1.35x**  | **1.20x** | **1.23x**  | **1.25x** |
>
> > R3-2-Table2: Experimental results on Vicuna-13B-v1.3 (Greedy Decoding, FP16 Precision)
>
> | Methods   |  Writing  | Roleplay  | Reasoning |   Math    |  Coding   | Extraction |   Stem    | Humanities |  Overall  |
> | --------- | :-------: | :-------: | :-------: | :-------: | :-------: | :--------: | :-------: | :--------: | :-------: |
> | Lookahead |   1.08x   |   1.17x   |   1.10x   |   1.19x   |   1.15x   |   1.16x    |   1.09x   |   1.14x    |   1.14x   |
> | SWIFT     | **1.24x** | **1.31x** | **1.29x** | **1.24x** | **1.35x** | **1.45x**  | **1.28x** | **1.30x**  | **1.31x** |
>
> These results demonstrate that SWIFT **outperforms** Lookahead across all MT-Bench subtasks, achieving substantial gains in overall efficiency. Additionally, we would like to note that each subtask in MT-bench is limited to 10 instances. In real-world LLM chat applications, as we addressed in the prior response, by caching optimal settings and continually optimizing on similar input instances, the efficiency of SWIFT could be further enhanced.
>
> [1] Break the Sequential Dependency of LLM Inference Using Lookahead Decoding. Fu et.al. ICML 2024.
>
> ------
>
> **To sum up:**
>
> The above clarifications and additional evidence underscore SWIFT’s contributions as the **state-of-the-art plug-and-play SD method**. It not only provides an orthogonal complement to Lookahead but also demonstrates superior efficiency gains across various benchmarks. If you acknowledge this claim, we believe the above reasons for negative scores have been well addressed and we hope that you will **reconsider the basis for the current score**. If you have any further concerns about the above claim or additional reasons for maintaining the negative scores, please let us know.  We are eager to have a deep discussion with you.

---

> > ### Comment · Reviewer_LGVh · 2024-11-25
> >
> > Thank you for your detailed response. In Figure 6, you demonstrate that SWIFT can dynamically optimize layer skip configurations. However, this seems to rely on an assumption that consecutive instructions belong to the same task type. In real-world application, with a massive number of user instructions, the diversity is much higher, and this assumption often does not hold. Therefore, I’m curious: if instances in MT-Bench are processed in a random order, what would the speedup ratio be? Additionally, I would still like to see a comparison of speedup between Lookahead and SWIFT on MT-Bench using Llama-3 as the backbone.

---

> ### Author Response · Authors · 2024-11-25
> **Response to Reviewer LGVh**
>
> Thank you for your prompt feedback. Below, we provide additional experimental results and discussions to address your concerns.
>
> **1.Comparisons on MT-Bench in random order**
>
> We conducted additional experiments comparing SWIFT and Lookahead using a random order of MT-Bench, as shown below:
>
> > R3-3-Table1: Experimental Results on Vicuna-v1.3 (Greedy Decoding, FP16 Precision)
>
> | Methods   | Vicuna-7B | Vicuna-13B | Vicuna-33B |
> | --------- | :-------: | :--------: | :--------: |
> | Lookahead |   1.13x   |   1.14x    |   1.13x    |
> | SWIFT     | **1.20x** | **1.27x**  | **1.35x**  |
>
> In this experimental setting, SWIFT continues to demonstrate **superior efficiency** over Lookahead across all model sizes. Moreover, as the model size increases, SWIFT’s overall speedup **improves consistently**, aligning with the trends observed in Figure 8 of our main paper.
>
> We acknowledge your concern that "in real-world applications, with a massive number of user instructions, the diversity of instances is much higher." However,  as we addressed in our prior response, most user instructions can be categorized into similar types based on their intent, such as reasoning, writing, coding, QA, etc. (e.g., the 8 subtasks in MT-Bench). Therefore, a potential application of SWIFT in these scenarios could involve **caching optimized layer configurations for similar data types** and retrieving them when processing corresponding instances. This approach would likely **further enhance** SWIFT’s efficiency beyond the results shown here.
>
> That is to say, the speedup results in `R3-3-Table1` represent a **lower bound** for SWIFT’s efficiency, as they rely solely on instance-specific optimization. Even under these restricted conditions, SWIFT outperforms the prior plug-and-play method, Lookahead. We sincerely hope that our response could provide you a better understanding of SWIFT's value and potential for efficient LLM inference.
>
> **2.Additional comparisons of LLaMA-3 on MT-Bench**
>
> We also conducted evaluations on MT-Bench using LLaMA-3, as presented below:
>
> > R3-3-Table2: Experimental results on LLaMA-3-8B-Instruct (Greedy Decoding, FP16 Precision)
>
> | Methods   |  Writing  | Roleplay  | Reasoning |   Math    |  Coding   | Extraction |   Stem    | Humanities |  Overall  |
> | --------- | :-------: | :-------: | :-------: | :-------: | :-------: | :--------: | :-------: | :--------: | :-------: |
> | Lookahead |   1.05x   |   1.14x   |   1.10x   |   1.23x   |   1.15x   |   1.16x    |   1.11x   |   1.15x    |   1.14x   |
> | SWIFT     | **1.24x** | **1.26x** | **1.24x** | **1.21x** | **1.26x** | **1.30x**  | **1.22x** | **1.19x**  | **1.24x** |
>
> > R3-3-Table3: Experimental results on LLaMA-3-70B-Instruct (Greedy Decoding, FP16 Precision)
>
> | Methods   |  Writing  | Roleplay  | Reasoning |   Math    |  Coding   | Extraction |   Stem    | Humanities |  Overall  |
> | --------- | :-------: | :-------: | :-------: | :-------: | :-------: | :--------: | :-------: | :--------: | :-------: |
> | Lookahead |   1.06x   |   1.15x   |   1.11x   |   1.22x   |   1.19x   |   1.14x    |   1.10x   |   1.14x    |   1.14x   |
> | SWIFT     | **1.31x** | **1.43x** | **1.36x** | **1.34x** | **1.44x** | **1.52x**  | **1.33x** | **1.37x**  | **1.39x** |
>
> These results confirm SWIFT’s superiority over Lookahead when LLaMA-3 serves as the backbone, consistent with our findings in `R3-2-Table1` and `R3-2-Table1`.
>
> ------
>
> We hope the above demonstrations and additional experiments comprehensively address your concerns. We deeply appreciate your inquiry about SWIFT's effectiveness in real-world LLM chat applications. We will incorporate these discussions into the revised manuscript and remain open to further feedback. Please feel free to reach out with any additional concerns.
>
> Thank you for your thoughtful review.

---

> > ### Comment · Reviewer_LGVh · 2024-11-25
> >
> > Thanks a lot. I have no more concerns and I raise my score to 5.

---

### Official Review · Reviewer_tWD9 · 2024-11-03

**Soundness:** 3
**Presentation:** 3
**Contribution:** 2
**Rating:** 6
**Confidence:** 3

**Summary:**

This paper aims to improve speculative decoding (SD) with a focus on eliminating the need for additional model parameters or extensive training to enable effective drafting in SD. In particular, the paper utilizes the same (target) model as the draft model by skipping a subset of model layers while generating draft tokens. Towards this, the paper proposes an SD method, namely SWIFT, that performs on-the-fly adaptive layer selection via an optimization phase to identify task-specific layers to skip. The optimization phase is followed by an inference acceleration phase that leverages the identified layers to perform skipping during drafting. During the inference acceleration phase, SWIFT additionally relies on 1) early stopping of the drafting process if (draft) model's confidence is not high enough; and 2) utilizing top-k predictions for each draft token position during parallel verification. The authors empirically validate the utility of SWIFT by showcasing 1.3-1.6x speed-up on CNN/DM, GSM8K, and TinyStories datasets.

**Strengths:**

1) The paper successfully demonstrates that the speculative decoding (SD) framework has the potential to speed up LLM inference even when one does not employ additional model parameters and task-specific training to support the drafting phase.

2) The paper makes two key observations about layer skipping during the drafting phase that highlights the need for adaptive (task-specific) selection of layers to skip during the drafting phase to maximize the benefit of layer skipping-based drafting approach. Subsequently, the paper proposes SWIFT - an effective SD approach that can identify a reasonable set of layers to skip for the underlying task with minimal training.

3) The paper further showcases the utility of leveraging the (draft) model's prediction confidence and top-k per-token predictions to improve the realized speed-up via SWIFT.

4) The paper is mostly well-written and conveys the key ideas in sufficient detail. The proposed ideas exhibit sufficient novelty over existing SD methods. The empirical results and in-depth empirical analysis highlight the gains realized by SWIFT over vanilla LLM inference.

**Weaknesses:**

1) There is room for improvement in the discussion of related prior work. Given that Elhoushi et al. 2024 also leverage layer skipping during the drafting phase, a detailed discussion of this work is warranted. Furthermore, the authors may also want to cite https://openreview.net/pdf?id=yUmJ483OB0.

2) The authors may want to make their empirical evaluation more comprehensive. Currently, the authors don't compare with existing approaches that rely on layer skipping during the drafting phase. Even though these existing methods might rely on extensive training, the authors should compare SWIFT with these methods. Such a comparison can highlight if there is any performance gap between these methods and their proposed plug-and-play approach.

3) The paper aims to eliminate the extensive training of existing layer skipping-based approaches via an efficient on-the-fly optimization phase. However, it's not clear if the existing methods can also perform well even when one limits the amount of offline training for these methods.

4) The authors repeatedly emphasize that their proposed method is a plug-and-play method. However, they don't seem to be evaluating their method in a dynamic setting where the underlying task (distribution) changes over time. In such a dynamic setting, would SWIFT have to interleave the optimization and acceleration phases? Would one still observe a good amount of speed up in such settings?

**Questions:**

Please see the weaknesses section above. In addition, please consider the following questions:

1) Looking at the ablation studies in Appendix D (Table 7), it appears that *dynamic verification* does not bring much value as the loss in overall speed-up is minimal when one excluded dynamic verification (1.560x to 1.541x). Could authors comment on this?

2) Do the speedup numbers in Table 2 take into account the optimization phase? If yes, how many LLM generations are performed to obtain the results in Table 2?

---

> ### Author Response · Authors · 2024-11-17
> **Response - 1**
>
> We sincerely appreciate your thoughtful and professional comments. We are delighted by your recognition of our main idea, from its motivation to the experimental validation of "the great potential of LLMs for self-acceleration without additional model parameters or task-specific training."  We are also encouraged by your acknowledgment of our two key empirical observations, as we believe basing our work on experimental evidence provides a strong foundation for further exploration.
>
> We are also grateful for your remarks on the paper’s clarity, noting that our key ideas are presented with sufficient detail and that SWIFT offers meaningful *novelty* compared to prior SD methods. Your recognition of these aspects motivates us to continue refining and advancing this line of research.
>
> In addition to your praise for the paper’s main contributions, we have carefully considered your constructive feedback and suggestions for clarification and additional experiments. We agree that these points enhance the robustness of our findings and further validate the main ideas without detracting from their significance.
>
> Below, we provide detailed, point-by-point responses to each of your comments.
>
> ***Q1: There is room for improvement in the discussion of related prior work. Given that Elhoushi et al. 2024 also leverage layer skipping during the drafting phase, a detailed discussion of this work is warranted. Furthermore, the authors may also want to cite Yang et al.***
>
> A1: Thank you for this insightful feedback. LayerSkip [1] explores an innovative approach to self-speculative decoding by implementing early-exit drafting, where the LLM generates drafts using only its earlier layers and then verifies these drafts with the full-parameter LLM. To support this process, LayerSkip **necessitates a time-consuming training process** involving layer dropout and early exit losses, which, while effective, demands significant computational resources for either pretraining or task-specific fine-tuning (as compared in R2). Moreover, this training process **modifies the original output distribution of the target LLM**, potentially impacting the consistency and reliability of LLM generation outputs.
>
> > Similarly, PPD [2] also explores early-exiting drafting; however, rather than relying on a single language modeling classifier from the final layer, PPD investigates classifiers trained for each individual layer.
>
> In comparison to LayerSkip [1], our proposed SWIFT selects intermediate layers of LLMs to skip *on the fly*, without requiring auxiliary models or additional training processes, making it a *plug-and-play* solution for accelerating LLM inference. Furthermore, SWIFT theoretically preserves the original output distribution of the target LLM, achieving a stable 1.3x-1.6x speedup without altering model behavior. We will integrate these points into a revised discussion on related work in our manuscript.
>
>
>
> [1] Layer Skip: Enabling Early Exit Inference and Self-Speculative Decoding. Elhoushi et.al. ACL 2024.
>
> [2] Predictive Pipelined Decoding: A Compute-Latency Trade-off for Exact LLM Decoding. Yang et.al. TMLR 2024.

---

> ### Author Response · Authors · 2024-11-17
> **Response - 2**
>
> ***Q2: Currently, the authors don't compare with existing approaches that rely on layer skipping during the drafting phase. Even though these existing methods might rely on extensive training, the authors should compare SWIFT with these methods. Such a comparison can highlight if there is any performance gap between these methods and their proposed plug-and-play approach.***
>
> A2: Thanks for this advice! We provide a comparison of SWIFT with LayerSkip [1] and Self-SD [2] below, which are the two most representative layer-skipping SD methods. We report the skip ratio ($r$), mean accepted tokens (*M*), and token acceptance rate ($\alpha$)  for comparison. The relationship among these three metrics and the expected wall-clock speedup is illustrated in Equation (6) of Appendix B.3.
>
> > R2-Table1: Experimental Results on LLaMA-2-13B, CNN/DM (Greedy Decoding, FP16 Precision)
>
> | Methods         | Plug-and-Play | Original Dist | $r$  | *M*  | $\alpha$ |  Speedup  |
> | --------------- | :-----------: | :-----------: | :--: | :--: | :------: | :-------: |
> | LayerSkip       |    **No**     |    **No**     | 0.80 | 2.42 |   0.64   |   1.64x   |
> | Self-SD         |      No       |      Yes      | 0.43 | 4.02 |   0.85   |   1.29x   |
> | Self-SD *w/ CA* |      No       |      Yes      | 0.43 | 5.69 |   0.98   |   1.52x   |
> | SWIFT           |      Yes      |      Yes      | 0.45 | 5.82 |   0.98   | **1.56x** |
>
> > *CA* refers to the Confidence-aware inference Acceleration strategy in Section 4.2. 'Original Dist' indicates whether the original distribution of the target LLM is altered.
> >
> > Note: We re-implemented LayerSkip using the huggingface version, which does not support KV cache reuse. Integrating KV cache reuse would likely improve LayerSkip's speedup to approximately 1.8x, as reported in its original paper.
>
> We compare SWIFT with each layer-skipping SD method below:
>
> - **Comparison with LayerSkip:** LayerSkip’s pretraining/finetuning process enables a more aggressive skip ratio ($r=0.8$), resulting in an average score of $M=2.42$ and $\alpha=0.64$. However, as noted in **R1**, this process modifies the ***original distribution*** of the target LLM, potentially reducing the reliability of its outputs. In contrast, SWIFT preserves the original distribution of the target LLM, while achieving a promising 1.56x speedup.
> - **Comparison with Self-SD:** Self-SD necessitates a time-intensive Bayesian Optimization process before inference (~7.5 hours for LLaMA-2-13B and ~20 hours for LLaMA-2-70B). In contrast, SWIFT introduces an on-the-fly optimization strategy, resulting in an approximate **200X** reduction in optimization latency while maintaining a 1.56x speedup. We further augmented Self-SD with our *Confidence-aware inference Acceleration strategy* (Self-SD *w/ CA*). Even compared to this augmented version, SWIFT achieves competitive speedups.
>
> To further illustrate SWIFT’s efficiency, we present a breakdown of the training and optimization costs for these methods:
>
> > R2-Table2: Comparison of training and optimization costs for Llama-2-13B
>
> | Methods              | LayerSkip                               | Self-SD                                                | SWIFT                                       |
> | -------------------- | :-------------------------------------- | :----------------------------------------------------- | :------------------------------------------ |
> | Training Cost        | 50k training steps with 64 A100 (80 GB) | 1000 Bayesian Optimization Iterations Before inference | N/A                                         |
> | Optimization Latency | -                                       | ~7.2 hours                                             | ~2 minutes (**200X** reduction$\downarrow$) |
>
> These comparisons clearly highlight SWIFT’s efficiency in both performance and training/optimization costs. We will incorporate these results and discussions into our revised manuscript. We sincerely appreciate your suggestion, which has helped us strengthen the comparative analysis and better illustrate SWIFT’s advantages.
>
>
>
> [1] Layer Skip: Enabling Early Exit Inference and Self-Speculative Decoding. Elhoushi et.al. ACL 2024.
>
> [2] Draft & Verify: Lossless Large Language Model Acceleration via Self-Speculative Decoding. Zhang et.al. ACL 2024.

---

> ### Author Response · Authors · 2024-11-17
> **Response - 3**
>
> ***Q3: The paper aims to eliminate the extensive training of existing layer skipping-based approaches via an efficient on-the-fly optimization phase. However, it's not clear if the existing methods can also perform well even when one limits the amount of offline training for these methods.***
>
> A3: Thank you for raising this point. Below, we provide a detailed comparison of optimization performance, focusing on Self-SD [1], an established layer-skipping SD approach, with varying amounts of optimization iterations.
>
> > R2-Table3: Experimental Results of Self-SD on LLaMA-2-13B, CNN/DM (Greedy Decoding, FP16 Precision)
>
> | #Bayesian_Opt | Optimization Time (s) | $r$  | *M*  | $\alpha$ | Speedup |
> | :-----------: | :-------------------: | :--: | :--: | :------: | :-----: |
> |       0       |           0           | 0.50 | 1.75 |   0.56   |  0.96x  |
> |      10       |          279          | 0.49 | 1.83 |   0.57   |  0.97x  |
> |      50       |         1474          | 0.49 | 1.80 |   0.61   |  1.02x  |
> |      100      |         2898          | 0.45 | 3.04 |   0.80   |  1.19x  |
> |      200      |         5517          | 0.48 | 3.47 |   0.84   |  1.24x  |
>
> As shown, Self-SD achieves minimal speedup improvement with fewer than 50 Bayesian optimization iterations (nearly equivalent to unified skipping, i.e., *#Bayesian Opt = 0*). At 100 iterations, Self-SD reaches a 1.19x speedup; however, its optimization latency is nearly **25 times** that of SWIFT (~1 hour).
>
> To compare SWIFT and Self-SD under similar optimization latencies, we conducted the following experiment:
>
> > R2-Table4: Experimental Results on LLaMA-2-13B, CNN/DM (Greedy Decoding, FP16 Precision)
>
> | Methods       | #Random_Opt | #Bayesian_Opt | Opt_Time (s) | $r$  | *M*  | $\alpha$ | Speedup |
> | ------------- | :---------: | :-----------: | :----------: | :--: | :--: | :------: | :-----: |
> | Self-SD       |      -      |       5       |     155      | 0.50 | 1.80 |   0.57   |  0.97x  |
> | Self-SD w/ CA |      -      |       5       |     155      | 0.50 | 2.07 |   0.86   |  1.17x  |
> | SWIFT         |     552     |      23       |     **116**      | 0.45 | 5.82 |   0.98   |  **1.56x**  |
>
> These results demonstrate SWIFT’s superiority over Self-SD in both optimization efficiency and speedup. Below, we analyze the reasons for this advantage (discussed in L168–L174 of our manuscript):
>
> - **Optimization Objective Granularity:** Self-SD calculates its optimization objective at a multi-sample level, requiring sequential decoding of all selected training samples (e.g., 8 samples with 32 tokens each) for every iteration to optimize Equation (1). In contrast, SWIFT adopts a **step-level optimization objective**, optimizing the layer set dynamically at each decoding step.
> - **Bayesian Optimization Complexity:** The computational complexity of Bayesian optimization increases significantly with the number of iterations. SWIFT mitigates this burden by combining **random search** with **interval Bayesian optimization**, accelerating convergence of the optimization process while reducing computational overhead.
>
> To further explore optimization trade-offs, we reduce Self-SD’s sequential optimization demand to 1 sample with 8 tokens, allowing for more Bayesian optimization iterations under similar latency. The results are summarized below:
>
> > R2-Table5: Experimental Results on LLaMA-2-13B, CNN/DM (Greedy Decoding, FP16 Precision)
>
> | Methods       | #Random_Opt | #Bayesian_Opt | Opt_Time (s) | $r$  | *M*  | $\alpha$ | Speedup |
> | ------------- | :---------: | :-----------: | :----------: | :--: | :--: | :------: | :-----: |
> | Self-SD       |      -      |      30       |     199      | 0.45 | 2.08 |   0.70   |  1.04x  |
> | Self-SD w/ CA |      -      |      30       |     199      | 0.45 | 2.44 |   0.93   |  1.22x  |
> | SWIFT         |     552     |      23       |     **116**      | 0.45 | 5.82 |   0.98   |  **1.56x**  |
>
> Even with optimized settings, SWIFT achieves significantly better speedup and efficiency compared to Self-SD, demonstrating the superiority of our proposed strategies.
>
> We appreciate your insightful question and will incorporate these results and discussions into the revised manuscript.
>
>
>
> [1] Draft & Verify: Lossless Large Language Model Acceleration via Self-Speculative Decoding. Zhang et.al. ACL 2024.

---

> ### Author Response · Authors · 2024-11-17
> **Response - 4**
>
> ***Q4: However, they don't seem to be evaluating their method in a dynamic setting where the underlying task (distribution) changes over time. In such a dynamic setting, would SWIFT have to interleave the optimization and acceleration phases? Would one still observe a good amount of speed up in such settings?***
>
> A4:  Thank you for this insightful question. **If we understand your concern correctly, we have indeed validated SWIFT’s effectiveness in handling dynamic input data streams, as detailed in Lines 474–495 of our manuscript.** SWIFT incorporates a dynamic mechanism that adaptively triggers the optimization phase whenever the token acceptance rate falls below 0.93. This enables SWIFT to optimize on-the-fly for each domain during inference without requiring extensive pretraining.
>
> To evaluate its performance, we conducted experiments across various tasks—summarization, reasoning, instruction following, translation, and question answering. For each task, we sampled 500 instances from the respective test sets and concatenated them sequentially to form a dynamic input stream.
>
> **Results in Figure 7:**
>
> - SWIFT demonstrated strong adaptability across domains, achieving an ***average token acceptance rate of 96%*** and maintaining a consistent ***1.3x–1.6x speedup***.
> - In contrast, Self-SD was highly sensitive to domain shifts, with its average token acceptance rate dropping from 92% to 68%. This decline resulted in a severe reduction in speedup, falling from 1.33x to an average of 1.05x under domain shifts.
>
> These findings highlight SWIFT’s ability to dynamically adapt to changing task distributions while maintaining efficiency. We will further elaborate on this dynamic evaluation in the revised manuscript.
>
> ***Q5: Looking at the ablation studies in Appendix D (Table 7), it appears that dynamic verification does not bring much value as the loss in overall speed-up is minimal when one excluded dynamic verification (1.560x to 1.541x). Could authors comment on this?***
>
> A5: Thank you for pointing this out! Upon review, we identified a numerical typo in our manuscript. Specifically, '*dynamic ver*' refers to the confidence-based top-k draft candidate extension in SWIFT’s inference strategy. Excluding this mechanism results in a vanilla verification strategy similar to Self-SD, leading to a more substantial reduction in speedup—from 1.56x to 1.34x, not 1.541x as previously reported.
>
> We provide the corrected comparison results below:
>
> > R2-Table6: Ablation Results on LLaMA-2-13B, CNN/DM (Greedy Decoding, FP16 Precision)
>
> | Methods                 | $r$  | *M*  | $\alpha$ | Speedup |
> | ----------------------- | :--: | :--: | :------: | :-----: |
> | Self-SD                 | 0.43 | 4.02 |   0.85   |  1.29x  |
> | SWIFT *w/o dynamic ver* | 0.45 | 4.39 |   0.90   |  1.34x  |
> | SWIFT                   | 0.45 | 5.82 |   0.98   |  **1.56x**  |
>
> These corrected figures demonstrate that dynamic verification meaningfully improves both the token acceptance rate ($\alpha$) and speedup, underscoring its value in our inference strategy. We will ensure this correction is accurately reflected in the revised manuscript.
>
> Thank you again for your careful and thorough review. Your detailed feedback has helped us identify and address this issue, and we greatly appreciate the time and effort you put into evaluating our work. We hope that this correction clarifies our findings and ensures that this oversight does not negatively influence your evaluation of the manuscript.

---

> ### Author Response · Authors · 2024-11-17
> **Response - 5**
>
> ***Q6: Do the speedup numbers in Table 2 take into account the optimization phase? If yes, how many LLM generations are performed to obtain the results in Table 2?***
>
> A6: Yes, the speedup numbers in Table 2 reflect the overall wall-clock speedup, incorporating the latencies of both the optimization and acceleration phases for all evaluated methods. As described in Section 5.1, we randomly sampled 1,000 instances from the test set for each dataset, following the setup in Self-SD [1]. The maximum generation lengths for CNN/DM, GSM8K, and TinyStories were set to 64, 64, and 128 tokens, respectively.
>
> To further illustrate SWIFT’s efficiency, we provide a detailed prefilling analysis of its separate modules in Figure 6. This analysis shows that the optimization phase contributes minimally to the total inference latency, occupying just $\textbf{0.8}$\% of the total runtime. Specifically, the optimization phase concludes early in the process (by instance index 10), with the draft model achieving a satisfactory token acceptance rate of 0.98. Subsequently, SWIFT transitions to the acceleration phase.
>
> We report two key metrics in Figure 6 to clarify SWIFT’s efficiency:
>
> - **Overall Speedup:** Reflects the total wall-clock speedup, including both the optimization and acceleration phases.
> - **Instance Speedup:** Captures the speedup achieved for each individual instance.
>
> The results demonstrate that SWIFT’s overall speedup progressively increases as more tokens are generated, eventually converging toward the average instance speedup. This dynamic highlights a key feature of SWIFT: ***its efficiency scales with increasing input length and the number of instances***, making it particularly advantageous in large-scale inference scenarios.
>
> [1] Draft & Verify: Lossless Large Language Model Acceleration via Self-Speculative Decoding. Zhang et.al. ACL 2024.
>
>
> **To sum up:**
>
>
> In the discussion above, we have primarily clarified some misunderstandings (**A4, A6**), added experiments based on your interests (**A2, A3**), and conducted comparisons with prior work (**A1**). We sincerely thank you for your thoughtful suggestions and detailed feedback, which we have integrated into our paper without altering its original content. We hope our responses have effectively addressed your concerns.
>
> If you have any further questions or additional concerns, please feel free to discuss them with us. Additionally, we kindly request that you reconsider your rating in light of our responses. From our understanding, you hold a positive view of our work, and we believe the suggestions you raised have been appropriately addressed within the revised manuscript.

---

> > ### Comment · Reviewer_tWD9 · 2024-11-27
> > **Thank you for the detailed response**
> >
> > I thank the authors for their comprehensive response. Most of my questions and concerns are resolved. Adding new results (A2, A3) will further strengthen the submission. I have a couple of remaining questions:
> >
> > 1. In A3, why did the authors not perform a comparison with *LayerSkip with a limited training budget*?
> > 2. Do the authors expect to continue to observe good speedup in a dynamic setting when the change in data distribution is faster than what is considered in Figure 7? In real systems serving mixed traffic, is it common to have a large number of requests (~500) from a single task appear together?

---

> ### Author Response · Authors · 2024-11-23
> **Follow-Up: Seeking Further Feedback**
>
> Dear Reviewer, I hope you're doing well. Following up on our recent exchange regarding this paper, I wanted to check if there are any further concerns or feedback from your side. Your insights are invaluable to us, and we're keen to address any remaining issues.

---

> ### Author Response · Authors · 2024-11-25
> **Kindly Reminder on the Discussion Period**
>
> Dear Reviewer, I hope this message finds you well. As the discussion period is nearing its end with **only two days remaining**, I wanted to ensure we have addressed all your concerns satisfactorily. If there are any additional points or feedback you'd like us to consider, please let us know. Your insights are invaluable to us, and we’re eager to address any remaining issues to improve our work.
>
> Thank you for your time and effort in reviewing our paper.

---

> ### Author Response · Authors · 2024-11-27
> **Response to Reviewer tWD9**
>
> Thanks for your response. We are glad to hear that most of your questions and concerns have been addressed. And we appreciate your aknowledgement that the addtional results (A2, A3) further strengthen our submission. Below, we provide further clarifications to address your follow-up concerns.
>
> ***Q7: In A3, why did the authors not perform a comparison with LayerSkip with a limited training budget?***
>
> A7: Thank you for your inquiry. As we mentioned in Q2, training LayerSkip on LLaMA-2-13B requires **50k training steps with 64 A100s (80 GB)**, which involves significant computational resources. Given this large demand, we did not anticipate LayerSkip achieving an effective speedup with a comparative optimization latency on par with SWIFT (within 2 minutes). We deeply value the contributions of LayerSkip and the insights it provides to layer-skipping SD research as a training-required method. We note that our proposed SWIFT complements their efforts by investigating plug-and-play SD with layer skipping.
>
>
>
> ***Q8: Do the authors expect to continue to observe good speedup in a dynamic setting when the change in data distribution is faster than what is considered in Figure 7? In real systems serving mixed traffic, is it common to have a large number of requests (~500) from a single task appear together?***
>
> A8: Thanks for your further inquiry of SWIFT's effectiveness in real systems serving mixed traffic. This is a quite good point! To address this, we conducted additional experiments using MT-Bench [1], a widely adopted multi-turn benchmark with 8 subtasks (10 instances each). To simulate mixed traffic, we randomized the instance order in MT-Bench, which reflects a more dynamic real-world setting. The results are shown below:
>
> > R2-2-Table1: Experimental Results on Vicuna-v1.3 (Greedy Decoding, FP16 Precision)
>
> | Methods   | Vicuna-7B | Vicuna-13B | Vicuna-33B |
> | :-------- | :-------- | :--------- | :--------- |
> | Lookahead | 1.13x     | 1.14x      | 1.13x      |
> | SWIFT     | **1.20x** | **1.27x**  | **1.35x**  |
>
> In this experimental setting, SWIFT continues to demonstrate **superior efficiency** over Lookahead across all model sizes. Notably, as the model size increases, SWIFT’s overall speedup **improves consistently**, aligning with the trends observed in Figure 8 of our main paper.
>
> We also note that in LLM serving scenarios, most user instructions can be categorized into similar types based on their intent, such as reasoning, writing, coding, QA, etc. (e.g., the 8 subtasks in MT-Bench). Therefore, a potential application of SWIFT in these scenarios could involve **caching optimized layer configurations for similar data types** and retrieving them when processing corresponding instances. This approach would likely **further enhance** SWIFT’s efficiency beyond the results shown here.
>
> For further context on this point, we refer to a detailed discussion in our last response to Reviewer LGVh (R3).
>
> [1] Judging LLM-as-a-Judge with MT-Bench and Chatbot Arena. Zheng et al. NeurIPS 2023 Datasets and Benchmarks.
>
> ------
>
> We hope the additional experiments and clarifications comprehensively address your concerns. We deeply appreciate your thoughtful inquiry into SWIFT's effectiveness in real-world mixed traffic scenarios. These discussions will be incorporated into the final manuscript, and we remain open to any further feedback. Please feel free to reach out with any additional questions.
>
> Once again, we sincerely thank you for your thoughtful suggestions and detailed feedback, which helps us further strengthen our manuscript.

---

> > ### Comment · Reviewer_tWD9 · 2024-11-28
> > **Raised score**
> >
> > Thank you for providing additional experiments in response to my questions. I found the response adequate and have increased my score to 6.

---

> > > ### Author Response · Authors · 2024-11-29
> > >
> > > Thank you for your thoughtful feedback and revisiting your evaluation of our work. We sincerely appreciate your dedication and hard work throughout the review process. Wishing you a pleasant day ahead.

---

### Official Review · Reviewer_rnoa · 2024-11-08

**Soundness:** 3
**Presentation:** 2
**Contribution:** 3
**Rating:** 8
**Confidence:** 5

**Summary:**

By adaptively skipping intermediate layers during inference, SWIFT improves speedups of LLMs without compromising the quality of generation. The method integrates a Bayesian optimization-based layer selection mechanism to adapt to task-specific requirements dynamically.

**Strengths:**

While the concept of layer-skipping is not really novel, its use of bayesian optimization can be a good idea for self-SD.

**Weaknesses:**

1. The reward design and its stability under distributional changes need more explanation. Open discussion with concurrent work, such as "A Unified Framework for Speculative Decoding with Multiple Drafters as a Bandit (Submitted at ICLR'25; https://openreview.net/forum?id=5haYLrlyGj)", could enhance understanding of these challenges. While the primary focus is different, the insight of using bandit approach is quite similar to this paper. And I recommend the authors to put the discussions for the assumption and extensions of Bayesian optimization for layer skipping inspired by this work.


2. More discussions on related work, such as Kim et al. (2024), Stern et al. (2018), and Gloeckle et al. (2024) on pretrained blockwise parallel language models, would position the contribution better within the existing literature. Because both papers are also a parallel line of work for self-speculative decoding, while they use the non-autoregressive heads instead.

- Gloeckle et al. (2024),  Better & Faster Large Language Models via Multi-token Prediction.
- Stern et al. (2024), Blockwise Parallel Decoding for Deep Autoregressive Models
- Kim et al. (2024), Accelerating Blockwise Parallel Language Models with Draft Refinement. (https://openreview.net/forum?id=KT6F5Sw0eg)

**Questions:**

1. How does SWIFT handle non-stationary input distributions during Bayesian optimization?

2. Could the authors provide insights into how SWIFT performs under extreme token count variations or highly domain-specific tasks?

---

> ### Author Response · Authors · 2024-11-21
> **Response - 1**
>
> We sincerely thank Reviewer rnoa for your thoughtful review and are delighted that you recognize the value of our application of Bayesian Optimization in advancing plug-and-play SD research. Below, we provide detailed responses to each of your comments.
>
>
>
> ***Q1: Open discussion with concurrent work [1] could enhance understanding of these challenges. While the primary focus is different, the insight of using bandit approach is quite similar to this paper. And More discussions on related work [2,3,4]. These papers are also a parallel line of work for self-speculative decoding, while they use the non-autoregressive heads instead.***
>
> A1: Thank you for your insightful suggestions and for pointing out relevant concurrent work. We appreciate the opportunity to discuss these connections.
>
> [1] introduces MetaSD, an advanced Speculative Decoding (SD) method that integrates multiple specialized drafters into the target LLM and employs a multi-armed bandit sampling strategy to dynamically select the optimal drafter during inference. While the optimization target and implementation strategy in [1] differ from our work, both approaches share a common goal of dynamically optimizing the drafter configuration for SD, showcasing a promising direction for advancing SD research. Specifically, [1] employs a multi-armed bandit mechanism to switch between a fixed number of $k$ specialized drafters fine-tuned during the training stage. In contrast, **SWIFT’s search space is significantly larger**, as it involves determining the layer-skipping index for a given skip ratio, which has a combinatorial complexity of $\binom{L}{rL}$ as noted in L299-300. This challenge necessitated our use of Bayesian Optimization. We are keen to explore additional optimization strategies, including the bandit mechanism, in future work.
>
> We also appreciate your recommendation to discuss [2, 3, 4], which represent an exciting parallel line of work focusing on *non-autoregressive drafting* strategies. These methods integrate multiple draft heads into the target LLM, enabling parallel generation of draft tokens at each decoding step. Notably, [4] builds on the BPD paradigm introduced in [2], accelerating inference by refining block drafts with task-independent n-grams and lightweight rescorers using smaller LMs. While these approaches require extensive training of draft models, **SWIFT complements their efforts by exploring a plug-and-play SD paradigm** that does not rely on auxiliary models or additional training, offering a more flexible and practical solution.
>
> We will incorporate discussions of [1, 2, 3, 4] into our revised manuscript to provide a broader context for our work and highlight its position within the current landscape of SD research.
>
>
>
> [1] A Unified Framework for Speculative Decoding with Multiple Drafters as a Bandit.
>
> [2] Blockwise Parallel Decoding for Deep Autoregressive Models. Stern et al. NIPS 2018.
>
> [3] Better & Faster Large Language Models via Multi-token Prediction. Gloeckle et al. ICML 2024.
>
> [4] Accelerating Blockwise Parallel Language Models with Draft Refinement. Kim et al. NIPS 2024.

---

> ### Author Response · Authors · 2024-11-21
> **Response - 2**
>
> ***Q2: How does SWIFT handle non-stationary input distributions during Bayesian optimization?***
>
> A2: Thank you for your question regarding SWIFT’s handling of non-stationary input distributions during Bayesian optimization.
>
> **SWIFT is specifically designed to handle dynamic input data streams, as discussed in Lines 474–495 of our manuscript.** It employs a dynamic mechanism that adaptively triggers the optimization phase whenever the token acceptance rate falls below 0.93. This ensures that SWIFT can optimize on-the-fly for each domain during inference, without relying on extensive pretraining.
>
> To assess its performance, we conducted experiments on various tasks, including summarization, reasoning, instruction following, translation, and question answering. For each task, we sampled 500 instances from the respective test sets and concatenated them sequentially to create a dynamic input stream simulating domain shifts.
>
> **Key Results (Figure 7):**
>
> - SWIFT exhibited strong adaptability across domains, achieving an ***average token acceptance rate of 96%*** and maintaining consistent ***speedups of 1.3x–1.6x***.
> - By comparison, Self-SD struggled with domain shifts, showing a significant drop in the average token acceptance rate from 92% to 68%. This led to a sharp reduction in speedup, declining from 1.33x to an average of 1.05x under domain-shifted conditions.
>
> These results underscore SWIFT’s ability to dynamically adapt to non-stationary task distributions while maintaining efficiency and performance. In the revised manuscript, we will expand on this dynamic evaluation to provide further clarity and detail.
>
>
>
> ***Q3: Could the authors provide insights into how SWIFT performs under extreme token count variations or highly domain-specific tasks?***
>
> A3: Thank you for your insightful question! As highlighted in Section 4, SWIFT is designed to ***dynamically optimize its acceleration performance*** by adapting to the characteristics of the current data stream. A detailed analysis of its acceleration performance is provided in Figure 6, which demonstrates that SWIFT’s overall speedup progressively increases as more tokens are generated, ultimately converging toward the average instance speedup. This behavior underscores one of SWIFT’s key features: ***its efficiency scales with increasing input length and the number of instances***.
>
> This dynamic optimization mechanism also makes SWIFT particularly effective for ***highly domain-specific tasks*** involving large volumes of homogeneous data, such as those found in specific test sets or real-world industrial applications. In such scenarios, SWIFT can continuously refine its skipped layer configuration on the fly, enabling it to approach the optimal configuration for the domain-specific data.
>
> We greatly appreciate your feedback and hope this explanation clarifies SWIFT’s adaptability and versatility under domain-specific conditions. Let us know if further details are needed!
>
>
>
> **To sum up:**
>
> In the discussion above, we have provided detailed comparisons with prior work (**A1**) and clarified key features and strengths of SWIFT in response to your inquiries (**A2, A3**). We sincerely appreciate your thoughtful suggestions and detailed feedback, which we will incorporate into our paper while preserving its original contributions.
>
> We hope our responses have effectively addressed your concerns. Should you have any additional questions or further feedback, we would be happy to continue the discussion. Additionally, we kindly ask you to **reconsider your rating** in light of our responses. Based on your comments, we understand that you have a positive view of our work, and we believe the points you raised have been thoroughly addressed in our responses.
>
> Thank you again for your valuable input, and we look forward to further discussions.

---

> > ### Comment · Reviewer_rnoa · 2024-11-21
> >
> > Thank you for your detailed and thoughtful responses. I appreciate the effort to address each of my comments thoroughly and to integrate discussions of related work and clarifications on SWIFT's design. The added context and analysis significantly enhance the manuscript and align it well with the ICLR bar. I look forward to seeing the revised version incorporating the suggested points. I am confident in the value and contributions of this work and will revise my score to 8.

---

### Author Response · Authors · 2024-11-23
**General Response**

For clarity and simplicity, we will refer to Reviewers rnoa, tWD9, LGVh, and vnfL as R1, R2, R3, and R4, respectively, in the following response.

We sincerely thank all reviewers for their thoughtful and constructive feedback. We are encouraged by their recognition of the key contributions and strengths of our work.

In particular, we appreciate the acknowledgment of our empirical studies for making *crucial* observations on layer-skipping SD (**R2**), finding our methodology *insightful* for advancing SD research (**R1**), and appreciating our efforts to *successfully* demonstrate the great potential of LLMs for self-acceleration *without additional model parameters or task-specific training* (**R2, R3, R4**). Furthermore, we are pleased that our plug-and-play SWIFT method is viewed as *widely applicable* to most large language models (**R3**), that our approach is recognized for its *novelty* compared to existing SD methods (**R2**), and that the efficiency *superiority* of SWIFT over vanilla LLM inference is well acknowledged (**R2**). We also appreciate the reviewers’ comments noting that our paper is *generally well-written* (**R2, R4**), flows *smoothly* (**R4**), and conveys key ideas in *sufficient detail* (**R2**).

We have carefully addressed each individual comment provided by the reviewers and believe we have successfully responded to most of their concerns. In our revised manuscript, we have incorporated the suggested experiments, additional discussions, and relevant updates to further strengthen our work. Below, we summarize the core contributions of our study, the updates to our experiments, and the in-depth discussions included in our revision.

------

**Core Contributions of Our Work**

1. **Empirical Investigation**: We conducted an in-depth empirical analysis of LLM acceleration via layer sparsity, revealing the potential for LLM self-acceleration via layer sparsity and its task-specific nature, underscoring the necessity for adaptive self-speculative decoding during inference.
2. **Novel Framework**: we propose SWIFT, the first plug-and-play self-speculative decoding algorithm that dynamically optimizes the selection of skipped layers in the target LLM on the fly, enabling lossless acceleration of LLM inference across diverse input data streams.
3. **Complementary Efforts:** SWIFT represents a complementary research direction to existing plug-and-play SD methods. Its layer-skipping approach is orthogonal to Jacobi-based techniques like Lookahead Decoding, and combining the two could further amplify their collective efficiency.
4. **Experimental Results**: Through extensive experimentation across various models and tasks, we demonstrate that SWIFT consistently achieves a 1.3x-1.6x speedup without relying on auxiliary models or additional training, while theoretically guaranteeing the preservation of the generated text’s distribution.

------

**Updates of experimental results during Rebuttal**

- **`Appendix C.2`**: Added experimental results using LLaMA-2-70B-Chat and LLaMA-3-70B-series models, including both base LLMs and instruction-tuned variants.
- **`Appendix D.3`**: Detailed comparison with prior layer-skipping methods (e.g., LayerSkip[1] and Self-SD[2]), focusing on wall-clock speedups, training costs, and optimization latency.
- **`Appendix D.4`**: Analyzed the optimization burden of Self-SD[2] and compared its performance with SWIFT under similar optimization latency.
- **`Appendix D.1`**: Corrected the ablation study results.

**Updates of in-depth discussions during Rebuttal**

- **`Appendix D.5`**: Discussed the necessity and importance of plug-and-play SD methods for LLM acceleration.
- **`Appendix D.6`**: Elaborated on related work, including SD methods with early exiting and their distinctions from SWIFT.

------

We believe these additions and clarifications comprehensively address the reviewers' concerns and enhance the overall quality of our manuscript.  All revisions are highlighted in `magenta-colored` text for ease of reference. Our manuscript is updated on `Nov 23, AOE time`.

We look forward to the reviewers' favorable consideration and remain grateful for their valuable feedback.

[1] Layer Skip: Enabling Early Exit Inference and Self-Speculative Decoding. Elhoushi et.al. ACL 2024.

[2] Draft & Verify: Lossless Large Language Model Acceleration via Self-Speculative Decoding. Zhang et.al. ACL 2024.

---

### Meta-Review · Area_Chair_8857 · 2024-12-21

**Metareview:**

(a) Summary of Scientific Claims and Findings:

This paper introduces SWIFT, a plug-and-play self-speculative decoding (SD) method that accelerates large language model (LLM) inference by dynamically skipping intermediate layers during drafting. Layer-skipping is used as the compact draft model, avoiding additional training or parameters. A Bayesian optimization algorithm adaptively selects task-specific layers to skip during inference. SWIFT achieves speedups on various benchmarks without compromising text quality, which outperforms prior SD methods.

(b) Strengths of the Paper:

SWIFT does not require auxiliary models or additional training, enabling immediate deployment across various LLMs.

The authors introduce layer sparsity-based SD complements existing SD approaches and offers a practical alternative to training-heavy methods.

SWIFT is validated across multiple datasets and LLMs, with comparisons to existing methods like Self-SD and Lookahead.

(c) Weaknesses of the Paper and Missing Elements:

Limited evaluation against newer training-free methods, e.g., Lookahead, and methods with minimal training overhead, such as Mudusa and Eagle.

Needs more extensive testing in highly dynamic scenarios where task distribution changes rapidly.

Some reviewers noted limited speedup for well-optimized LLMs (e.g., Llama-3 series) compared to less efficient models.

(d) Decision and Rationale:

The paper has significant contributions, particularly its novel plug-and-play SD method. However, concerns about limited comparisons still remain. Positive reviewer feedback following the discussion phase suggests potential for acceptance.

**Additional Comments On Reviewer Discussion:**

The authors effectively addressed the reviewers’ most concerns, leading to an increase in the reviewers’ scores.

---

### Decision · Program_Chairs · 2025-01-22

Accept (Poster)